# Fairshare Data Pricing via Data Valuation for Large Language Models

**Luyang Zhang** *
Carnegie Mellon University
luyangz@andrew.cmu.edu

**Cathy Jiao** *
Carnegie Mellon University
cljiao@cs.cmu.edu

**Beibei Li** †
Carnegie Mellon University
beibeili@andrew.cmu.edu

**Chenyan Xiong** †
Carnegie Mellon University
cx@cs.cmu.edu

## Abstract

Training data is the backbone of large language models (LLMs), yet today's data markets often operate under exploitative pricing – sourcing data from marginalized groups with little pay or recognition. This paper introduces a theoretical framework for LLM data markets, modeling the strategic interactions between buyers (LLM builders) and sellers (human annotators). We begin with theoretical and empirical analysis showing how exploitative pricing drives high-quality sellers out of the market, degrading data quality and long-term model performance. Then we introduce *fairshare*, a pricing mechanism grounded in *data valuation* that quantifies each data's contribution. It aligns incentives by sustaining seller participation and optimizing utility for both buyers and sellers. Theoretically, we show that *fairshare* yields mutually optimal outcomes: maximizing long-term buyer utility and seller profit while sustaining market participation. Empirically when training open-source LLMs on complex NLP tasks, including math problems, medical diagnosis, and physical reasoning, *fairshare* boosts seller earnings and ensures a stable supply of high-quality data, while improving buyers' performance-per-dollar and long-term welfare. Our findings offer a concrete path toward fair, transparent, and economically sustainable data markets for LLM.

## 1 Introduction

High-quality training data is foundational to building effective and reliable large language models (LLMs). As LLMs take on increasingly complex tasks today – such as coding[1], reasoning [2], and AI4Science [3] – they rely heavily on carefully curated, human-annotated data. This growing demand has triggered a "generative data gold rush", with major tech companies racing to acquire training data, fueling the rise of a nascent AI data market [4]. In this market, AI firms create networks of short-term contract workers to generate data labels, resembling an Uber-like gig economy for data [4].

However, the current AI data market operates with limited oversight and is widely criticized for a lack of transparency and fairness in pricing [5–8]. Data prices are largely low and fail to reflect the quality or effort involved, threatening the sustainability and quality of data supply [9–11]. In particular, for data sellers, such as human annotators or content creators, the prevailing market routinely undervalues their labor, offering compensation that neglects the skill, effort, and downstream value of their contributions. These harms are especially concentrated in low-wage labor markets, where annotators often face overwork, underpayment, and exclusion from decision-making [12]. This reflects a broader ethical concern known as "AI parachuting", where developers extract data from marginalized

* Equal Contribution. †Co-Advisors.
39th Conference on Neural Information Processing Systems (NeurIPS 2025).

communities [13–15], tieing to wider debates on epistemic injustice and data colonialism [16–18]. As one civil rights advocate noted on 60 Minutes, "They don't pay well... they could pay whatever, and have whatever working conditions."[1]

Motivated by these issues, we present a fair pricing framework for the LLM training data market to promote equitable and sustainable generative AI ecosystems. Economic theory suggests that prices should reflect the value delivered to the buyer – signaling quality and alignment [19]. Guided by this, our framework introduces *fairshare* pricing based on established *data valuation* techniques for LLMs [20–23], which quantify each dataset's contribution to model performance. Buyers and sellers both have access to data valuation scores, which guide decision-making on both sides: buyers use them to select datasets under budget constraints to maximize utility (i.e., a standard measure of welfare or satisfaction [24, 25]), while sellers use them to set prices based on the anticipated demand from buyers. This shared access – enabled by our assumption of *information transparency* – supports procedural fairness [26], improves seller participation, and enhances overall data quality.

Our theoretical and empirical findings show that *fairshare* pricing offers clear advantages compared to existing methods. First, we show that existing exploitative pricing leads to a *lose-lose* outcome for the data market. For data buyers, underpaying data sellers might cut costs in the short term – but it comes at a cost of drive sellers away, shrinking the supply of high-quality training data. This weakens the data pipeline and limits model improvement, even as investments grow. In contrast, we show theoretically that *fairshare* pricing leads to a *win-win* outcome: sellers maximize profit while remaining engaged, and buyers secure long-term utility by maintaining access to high-quality data.

Second, we empirically validate our approach through simulations of buyer-seller interactions in data markets. We focus on training open-source LLMs on complex NLP tasks, including math problems [27], medical diagnosis [28], and physical reasoning [29]. Analyzing both pricing and valuation outcomes, we find that under *fairshare* pricing, buyers achieve higher model performance per dollar spent, making it particularly beneficial for those with limited budgets. In addition, our simulations of long-term market dynamics demonstrate that *fairshare* pricing encourages sustained seller participation, resulting in a stable and sufficient supply of training data over time compared to exploitative pricing. These findings show that our framework's data-valuation-based pricing not only improves short-term training efficiency, but also ensures the long-term viability of the data market.

Finally, to evaluate the robustness and method-agnostic applicability of our framework, we conduct an ablation study using a diverse set of *data valuation* methods (including BM25 [30], Infl$_{IP}$ [22], and Datainf[21]) – selected for their scalability and efficiency in LLM tasks. Across all variants, our *fairshare* pricing framework consistently delivers mutually beneficial results for both buyers and sellers in the LLM data market, confirming that its performance is not tied to any specific data valuation technique. This analysis underscores a key strength of our approach: buyers and sellers can flexibly choose valuation methods tailored to their downstream needs without compromising the incentive-aligned structure of the market, and demonstrates the broad applicability of our solution.

To summarize our contributions are as follows:

1. We present a novel theoretical framework to model the LLM training data market utilizing data valuation, economics of market, and game theory.

2. We propose a *fairshare* pricing mechanism that captures the strategic interplay between buyers and sellers. Both theory and empirical analyses show that exploitative pricing results in lose-lose outcomes, while *fairshare* promotes long-term stability and mutual benefits.

3. Our proposed pricing framework is highly generalizable across LLM models and tasks, and robust to the data valuation methods and utility measures used by market participants.

Our results demonstrate the versatility of our approach in data markets by aligning incentives, sustaining data quality, and maximizing long-term value. This highlights our *fairshare* mechanism as a guide for designing sustainable data procurement practices in public and private sectors.

---

[1]*CBS, 60 Minutes*

## 2 Related Work

**Data Market Design and Exploitative Pricing.** Recent studies on ML data markets focus on platform-based pricing and allocation mechanisms such as auctions and personalized pricing [31–35]. These frameworks typically aim to maximize profit or efficiency, assuming myopic buyers and fixed seller participation. They also consider *information transparency* ensures that sellers have access to the same *data valuation* scores used by buyers [36, 33]. The study was mainly conducted on classic machine learning models rather than large language models.

Concurrently, research has highlighted ethical concerns in LLM data acquisition, particularly around exploitative pricing. Annotation work – central to LLM training – is often outsourced to low-wage workers with minimal labor protections [9, 10, 37, 38], who are frequently exposed to harmful content [5, 12, 39]. Several studies document low pay, precarious labor, and lack of protections in global data supply chains [16–18]. Although these studies highlight systemic exploitation in data labor, their findings have yet to be integrated into formal models of LLM data pricing. Economic theory suggests that misaligned compensation undermines efficiency, discourages participation, and lowers data quality. In LLM contexts, this reduces data availability and degrades model performance [40–42], underscoring the need for value-aligned pricing mechanisms.

**Game-Theoretic Models of Pricing and Participation.** In many domains, pricing problems are modeled thorough game theory. Particularly, those involving decision-making are frequently modeled using *Stackelberg games* and *bilevel optimization* [43]. Although these frameworks have not yet been applied to LLM data market, they have been extensively used in contexts such as resource allocations [44], energy market [45, 46], supply chains [47], distributed systems [48]. In these settings, agents typically optimize revenue, efficiency, or utility while anticipating the strategic responses of others. In dynamic settings, these models extend to repeated interactions, where players balance immediate gains against long-term benefits [49, 43]. In repeated interactions, fairness perceptions are critical: studies show that perceived pricing unfairness can erode trust and reduce participation [50–53].

**Data Valuation Methods**: *Data valuation* estimates the contribution of training examples to model performance [54]. Examples include influence-function-based methods [55], which estimate data utility via model gradient computations. Variations such of this method enhance efficiency by approximating/bypassing the inverse-Hessian [56], or utilizing lower dimension model gradients (e.g., Datainf [21]), which balance efficiency and accuracy, and make them suitable for the LLM realm. Beyond the influence of loss functions, previous research [57] has examined data valuation with respect to fractional utility functions.

Data valuation approaches are effective for LLM training data selection [22, 23, 58] and other applications including toxicity detection [59], memorization analysis [60], training optimization [61], and and active sampling [62]. Shapley-based method is another direction [63–65, 36], such as CS-Shaply[66], In-run Shapley[67], DU-Shapley[68], etc. Simpler methods like BM25 [30] offer a model-agnostic baseline based on lexical similarity[69, 70]. While exact data value estimation is computationally costly at LLM scale, recent studies show that influence-based approximations correlate meaningfully with actual performance outcomes [71, 72]. For example, Jiao et al. [71] shows that influence-based methods have high correlation between their estimated data valuation scores and the oracle value, which is obtained through model re-training.

## 3 A Theoretical Framework of LLM Data Market

This section formalizes the LLM data market. We model buyer–seller interactions using economic utility theory and game theory, show that exploitative pricing leads to long-term market failure, and introduce the *fairshare* pricing strategy.

### 3.1 Data Market Definition

We formalize the LLM data market by modeling the decision-making processes of sellers and buyers in a non-cooperative *Stackelberg game* [35, 45, 48]. In this setup, sellers first set prices by anticipating buyer demand. Buyers then respond by selecting datasets to maximize their *utility*.[2] Each player

---

[2]*Utility* is defined as a general economic concept that measures an individual's welfare, benefit, or satisfaction [24, 25], for example, monetary value, model fairness, or other domain-specific benefits.

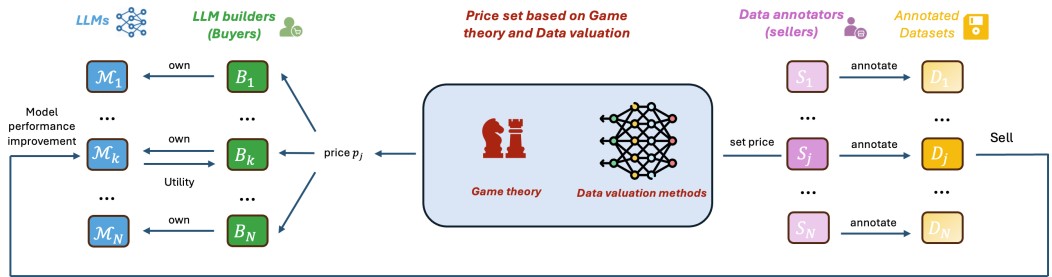

Figure 1: Illustration of the LLM data market, showing buyers (LLM builders) and sellers (annotators) interacting via a pricing mechanism based on game theory and data valuation.

makes decisions based solely on the current state of the market, without considering future updates or interactions.

Notably, our choice of a non-cooperative *Stackelberg* game fits the LLM data market's reality: many independent, self-interested transactions where binding grand coalitions are infeasible, such as gig-work platforms [18]. In contrast, cooperative bargaining targets few-party, contractible surplus splits, which is a different regime and question. In this non-cooperative game, we model fairness behaviorally via seller participation/exit that keeps the model tractable, and the leader–follower timing mirrors real price setting and demand response.

Following prior work [73], we assume *information transparency*: all participants observe each dataset's value. Practically, this can be estimated using *data valuation* methods that quantify its contribution to LLM performance. These valuation scores – potentially generated by the platform or a trusted third party – serve as common signals informing both pricing and purchasing decisions.

Next, we define key factors determining each player's decision-making. In practice, this can be estimated using *data valuation* methods that quantify its contribution to LLM performance. These scores – potentially generated by the platform or a trusted third party – serve as common signals guiding both pricing and purchasing decisions.

In the rest of this section, we formally define key factors determining each player's decision-making.

**Data Market Setup.** We begin by defining the players and the notion of *utility*. The market comprises a set of buyers $\{B_k\}_{k=1}^M$, each with an LLM $\mathcal{M}_k$, and a set of sellers $\{S_j\}_{j=1}^N$, each with a dataset $D_j$. Each buyer gains a certain level of economic *utility* from acquiring a new dataset based on a data value, which is captured by a *data valuation* function, $v_k : D \to \mathbb{R}_{\geq 0}$, measuring the marginal contribution of each dataset $D$ to the performance of buyer $B_k$'s LLM $\mathcal{M}_k$.

The value $v_k$ can be estimated using a *data valuation* method (e.g., *Influence Function* [55], DataInf [21], etc.). Once the *data valuation* score $v_k$ is estimated, the corresponding *utility* gain $u_k$ can be derived as a function of $v_k$. We use a task-specific mapping between data value $v_k$ and utility $u_k : v_k \to \mathbb{R}_{\geq 0}$. Appendix C lists common mappings between $u_k$ and $v_k$ in downstream tasks.

**Buyer's Decision-Making.** Each buyer $B_k$ selects a subset of datasets to acquire, maximizing the *net utility*, defined as the *utility* gain minus total acquisition cost.

$B_k$'s decision, represented by a binary vector $\boldsymbol{x}$ (where each entry is 1 if the corresponding dataset is selected, and 0 otherwise), depends on three components: (i) the current dataset prices, (ii) buyer's budget $b_k$, and (iii) the *utility* gain $u_k(\boldsymbol{x})$ from acquiring the selected datasets.

This *utility* gain reflects the downstream value, resulting from performance improvements in the buyer's model after training on the dataset bundle $\boldsymbol{x}$.[3] The *net utility* is defined as:

$$g_{k,N}\left(\boldsymbol{x}\right) := u_k\left(\boldsymbol{x}\right) - \boldsymbol{x}^T \boldsymbol{p}, \tag{1}$$

where $\boldsymbol{p} := [p_1, \ldots, p_N]$ is the price vector for current avaliable datasets.

---

[3]Datasets may exhibit dependencies. The set utility $u_k(\boldsymbol{x})$ is not necessarily additive over individuals.

Finally, $B_k$'s *purchasing problem* is formulated as selecting an optimal collection of datasets to maximize its *net utility*:

$$\tilde{\boldsymbol{x}}^{k,N} := \underset{\boldsymbol{x} \in \mathcal{X}_{k,N}}{\arg\max} \; g_{k,N}(\boldsymbol{x}), \quad \text{s.t.} \quad \mathcal{X}_{k,N} := \{\boldsymbol{x} \mid g_{k,N}(\boldsymbol{x}) \geq 0, \boldsymbol{x}^T \boldsymbol{p} \leq b_k\}, \tag{2}$$

where $\tilde{\boldsymbol{x}}^{k,N}$ is the optimal solution, and $\mathcal{X}_{k,N}$ includes all feasible solutions, $\boldsymbol{x}^T \boldsymbol{p} \leq b_k$ ensures that the total acquisition cost is within the budget $b_k$, and $g_{k,N}(\boldsymbol{x}) \geq 0$ ensures a non-negative *net utility*.

**Seller's Decision-Making.** When the seller $S_j$ offers dataset $D_j$, it sets a price to maximize its *net profit*, defined as (i) the anticipated sales from all buyers, minus (ii) a fixed cost to create the dataset. Anticipated sales from buyers are estimated by solving the previous buyer's *purchasing problem* (Equation (2)) known in the full transparent market. Formally, the seller's *net profit* function is:

$$r(p_j) := \sum_{k=1}^{M} \tilde{\boldsymbol{x}}_j^{k,N}(\boldsymbol{p}) p_j - c_j, \tag{3}$$

where $\tilde{\boldsymbol{x}}_j^{k,N}(\boldsymbol{p})$ is $j$-th entry of buyer $B_k$'s decision vector, indicating if $B_k$ purchases $D_j$ ($j$-th entry is 1) or not ($j$-th entry is 0), given $\boldsymbol{p}$ the prices of all datasets in the market; $c_j$ denotes the cost. The seller $S_j$ solves the following pricing problem:

$$p_j^* := \underset{p_j \in \mathcal{P}_{j,M}}{\arg\max} \; r(p_j), \quad \text{s.t.} \quad \mathcal{P}_{j,M} := \{p_j \in \mathbb{R}_+ \mid r(p_j) \geq 0\}, \tag{4}$$

where $p_j^*$ is the optimal price and $r(p_j) \geq 0$ ensures that seller $S_j$'s net profit must be non-negative. It is noted that we assume that selling data for annotator is profitable, i.e., $p_j^* \geq c_j$.

In the above formulation, buyer purchase decisions depend on prices and data value; seller pricing decisions depend on anticipated sales from buyers. Together, these decisions define the equilibrium dynamics of a transparent and incentive-aligned LLM data market. This formulation can be extended to a royalty-based scheme as presented in Appendix B.

## 3.2 Exploitative Pricing: A Lose-Lose Outcome

We now present a theoretical analysis showing that exploitative pricing in data markets is ultimately unsustainable and detrimental to all participants.

**Theoretical Setup.** We analyze a simplified dynamic setting involving a single buyer and a single seller over an infinite time horizon. At each time step $t$, both players decide whether to transact based on the proposed price. Importantly, to align with real-world market behavior, the seller's future participation depends on the history of past transaction prices.

We first introduce some assumptions on the seller behavior.

**Assumption 1** (*Declining Participation Probability*). When offered a price $p_t$ below the seller's ideal price $p_t^*$, the probability of seller's continued participation declines. This decline is captured by a strictly increasing function

$$\pi : (p_t, p_t^*) \rightarrow [0, 1], \text{ satisfying } \pi(0, p_t^*) = 0 \text{ and } \pi(p_t^*, p_t^*) = 1. \tag{5}$$

In addition, the probability that the seller $S$ remains active at time $T$ is $P_T := \prod_{t=0}^{T-1} \pi(p_t, p_t^*)$.

Assumption 1 models how the seller respond to exploitative pricing over time. It states that when offered a price below its ideal level, the seller become less likely to stay in the market, consistent with prior studies [74, 75]. For any exploitative pricing, the participation probability $P_T$ declines multiplicatively, leading to a sustained and irreversible drop in engagement.

To capture the degree of participation decline, we assume sellers respond sensitively to underpayment:

**Assumption 1.1** (*Sensitivity of Participation Is Lower-Bounded*). The Lipschitz continuity of $\pi$ over exploitative pricing is lower-bounded:

$$|\pi(p_{t,1}, p_t^*) - \pi(p_{t,2}, p_t^*)| \geq L |p_{t,1} - p_{t,2}|, \tag{6}$$

for some constant $L > 0$ and all exploitative pricing $p_{t,1}, p_{t,2} \in [0, p_t^*)$ for all $t$.

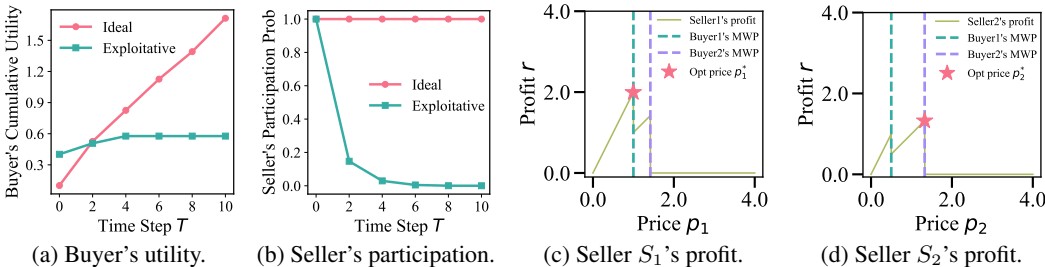

Figure 2: Buyer's cumulative utility and seller participation under ideal and exploitative pricing (2a,2b); Profits of sellers $S_1$ and $S_2$ with buyer $B_1$'s and $B_2$'s MWP over price (2c,2d).

We also introduce the following assumptions on buyers:

**Assumption 2** (*Discount Factor Is Lower-Bounded*). The buyer evaluates long-term utility using a discount factor $\delta$, satisfying:

$$\delta \geq \frac{1}{1 + L \min_{t \in [0,\infty)} \mathbb{E}[u_t - p_t^*]}, \forall t. \tag{7}$$

A higher discount factor $\delta$ indicates greater emphasis on future utility. Assumption 2 places a lower bound on $\delta$, ensuring that the buyer sufficiently values future gains when making decisions.

Assumption 2 assumes the buyer values both immediate and future utility gains from acquiring training data. Major LLM developers invest in large-scale data acquisition and model training in expectation of long-term gains in performance, deployment value, and commercial returns [76–79].

**Buyer's Objective.** With the previous setup, the buyer aims to maximize expected cumulative utility over an infinite horizon. Let $G(u_t, b_t)$ denote this value function, representing the buyer's expected total cumulative utility at time $t$, conditioned on current utility $u_t$ and budget $b_t$. Then this objective satisfies the following Bellman equation[80]:

$$G(u_t, b_t) = \max_{p_t \in [0,\infty)} \left[ \mathbb{E}\left[ u_t - p_t \right] + \delta \mathbb{E}\left[ \pi(p_t, p_t^*) G(u_{t+1}, b_{t+1}) \mid u_t, b_t \right] \right]. \tag{8}$$

This Bellman equation captures the buyer's central trade-off: offering an exploitative price $p_t$ increases immediate surplus $\mathbb{E}[u_t - p_t]$, but reduces future seller participation via a lower $\pi(p_t, p_t^*)$; conversely, setting a fairer price decreases short-term gain but sustains future transactions by increasing $\pi(p_t, p_t^*)$. With this insight, we then obtain the following result:

**Lemma 1** (*Inevitable Failure of Exploitative Pricing*). With Assumptions 1 to 2, any exploitative pricing (i.e., $p_t < p_t^*, \forall t$) will only maximize cumulative utility within a finite horizon – after which it is strictly suboptimal.

Lemma 1 reveals a fundamental limitation: any exploitative pricing strategy is only optimal for a finite time. Over time, it becomes strictly suboptimal, due to the declining seller participation. Thus, no exploitative strategy can maximize cumulative utility in the long run. (See Appendix D for the proof, shows that exploitative pricing yields a suboptimal value function.)

**Exploitative Pricing Leads to Lose-lose:** While exploitative pricing clearly reduces seller welfare, our results show it also harms long-term cumulative utility for buyers. Although buyers may benefit initially from lower costs, reduced seller participation quickly leads to a decline in data quality and availability. Over time, even well-resourced buyers face data shortages. This sets off a self-defeating cycle where short-term savings come at the expense of long-term model performance.

In addition to our theoretical findings, we run a simplified dynamic market simulation with one buyer and one seller, making sequential decisions. The utility of the dataset and buyer's budget varies randomly over time, and seller participation follows $\pi(p_t, p_t^*) = p_t/p_t^*$. Figures 2a and 2b provides consistent evidence that exploitative pricing causes rapid seller exit and an immediate shortage of training data. This dynamic closely resembles the classic "market for lemons" problem, where underpricing drives out high-quality supply, ultimately leading to market collapse [8].

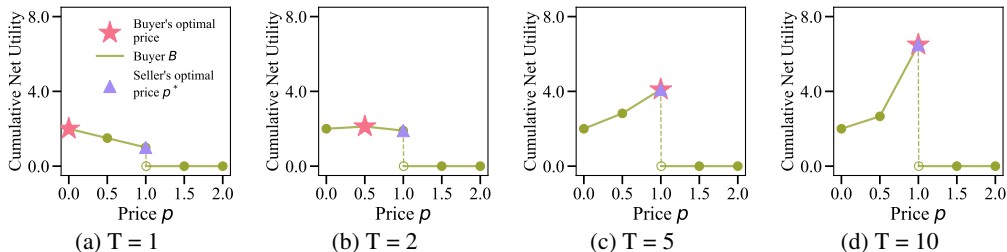

Figure 3: Analysis of the buyer's cumulative net utility as a function of the acquisition prices over time ($T = 1, 2, 5, 10$). Note: the market has one buyer and seller.

### 3.3 Fairshare Pricing: A Win-Win for Sellers and Buyers

The rest of this section presents the *fairshare* pricing in our LLM data market framework and show it yields a stable, *mutually optimal outcome* for both buyers and sellers.

**Seller-Side: Pricing Based on Buyers' Maximum Willingness to Pay.** When the seller $S_j$ prices its dataset $D_j$, it evaluates how each buyer's decision changes depending on whether $D_j$ is available at a given price. For each buyer $B_k$, the seller compares two scenarios: (1) the buyer's optimal dataset selection when $D_j$ is not part of the market, and (2) the buyer's new optimal selection if $D_j$ is included at price $p_j$. The buyer will purchase $D_j$ only if the dataset $D_j$ increases its *net utility* and remains within budget. Formally, let $\tilde{\boldsymbol{x}}^{k,N-1}$ be buyer $B_k$'s optimal decision without the presence of $D_j$. For any feasible decision $\boldsymbol{x} \in \mathcal{X}_{k,N-1}$, we define:

1. *Marginal utility* from adding $D_j$: $\Delta u_k(\boldsymbol{x}) := g_{k,N}(\boldsymbol{x} + e_j) - g_{k,N-1}(\tilde{\boldsymbol{x}}^{k,N-1})$, where $e_j$ is the unit vector indicating $D_j$ is selected and $p_j$ is set as zero, and

2. *Budget surplus* based on the prior decision: $\Delta b_k(\tilde{\boldsymbol{x}}^{k,N-1}) := b_k - (\tilde{\boldsymbol{x}}^{k,N-1})^T \boldsymbol{p}$.

A buyer's maximum willingness to pay (MWP) is defined as the highest price buyer $B_k$ is willing to pay – based on *marginal utility* – and able to pay – based on *budget surplus*:

$$\text{MWP}_k := \max_{\boldsymbol{x} \in \mathcal{X}_{k,N-1}} \{\min\{\Delta u_k(\boldsymbol{x})^+, \Delta b_k(\tilde{\boldsymbol{x}}^{k,N-1})\}\}, \tag{9}$$

where $\Delta u_k(\boldsymbol{x})^+ := \{\Delta u_k(\boldsymbol{x}), 0\}$ denotes the positive part of *marginal utility*, ensuring buyer $B_k$'s $\text{MWP}_k$ is non-negative – if the *marginal utility* of $D_j$ is negative, the buyer will not purchase it.

Then the seller's optimal price $p_j^*$ is the MWP that results the largest profit across buyers:

**Lemma 2** (*Characterization of Optimal Price $p_j^*$*). Seller $S_j$'s optimal price for $D_j$ is characterized as one of buyers' MWP (see Appendix D for proof):

$$p_j^* \in \cup_{k=1}^M \text{MWP}_k. \tag{10}$$

We see that optimal price $p_j^*$ is *fairshare* for the seller: it maximizes seller's profit while aligning with dataset's *utility* and buyer's budget.

As shown in Figures 2c and 2d, we simulate a data market with 2 sellers and 2 buyers, where players make one-shot decisions. Seller $S_1$ sets its price first, followed by seller $S_2$. Each plot shows each seller's profit function $r(p_j)$, with breakpoints at each buyer's MWP. Pricing above a buyer's MWP leads to a drop in sales. Thus, the seller's optimal price aligns with the MWP that yields maximum revenue.

**Buyer-Side: *Fairshare* Is Overall Optimal.** We now show that the *fairshare* price $p_j^*$, derived from seller optimization, is also optimal from the buyer's perspective. As established in Section 3.2, exploitative pricing leads to persistent seller exit. In contrast, *fairshare* pricing ensures full participation and maximizing the buyer's overall utility. To formalize this, we consider the same infinite-horizon setting in Section 3.2 with a single buyer and seller. Notably, the *fairshare* price is the ideal price for sellers as it maximizes its profit. Therefore, under *fairshare* pricing, the sellers will maintain sustained participation. Then, we obtain:

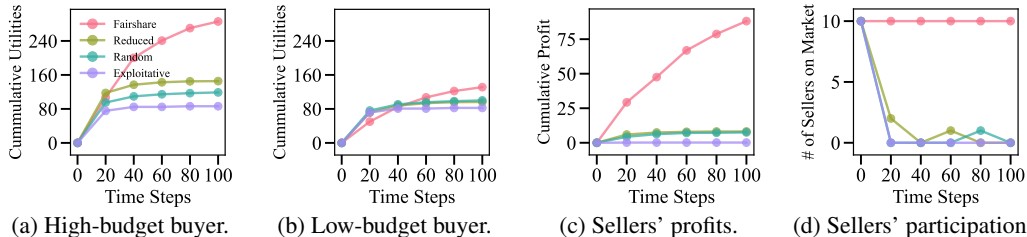

(a) High-budget buyer.  (b) Low-budget buyer.  (c) Sellers' profits.  (d) Sellers' participation.

Figure 4: (1) buyer's cumulative utilities with high- (Figure 4a) and low-budget buyer (Figure 4b), and (2) sellers' average cumulative profits (Figure 4c) and active seller numbers (Figure 4d) over 100 time periods. Pythia-1b; MedQA; Groups: (1) fairshare, (2) reduced, (3) random, and (4) exploitative.

**Lemma 3** (*The Optimal Price for Buyer Is Also $p_t^*$*). The *fairshare* price for the seller $S$ under LLM data framework is $p_t^* = \min\{u_t, b_t\}, \forall t$. With Assumptions 1 to 2, $p_t^*$ is the optimal pricing strategy that maximizes buyer's expected cumulative utility (Equation (8)).

In Figure 3, we develop a synthetic simulation illustrating that a buyer's optimal pricing strategy rapidly converges to the *fairshare* price. Consider a simulation with players receive constant utility $u_t = 2$ and budget $b_t = 1, \forall t$, with the discount factor $\delta = 0.95$, so the seller's optimal price remains $p_t^* = 1$. At each time step, the buyer selects a price that maximizes cumulative net utility. The buyer begins with a low, exploitative price at $T = 1$, but by $T = 5$, converges to the *fairshare* price and maintains it thereafter. This illustrates our core theoretical insight: fair pricing emerges as the optimal long-run strategy when buyers account for overall market sustainability.

Next, we also explore the role of the discount factor $\delta$ in our framework:

**Lemma 4** (*The Trade-Off Threshold Is Increasing as $\delta$ Decreases*). The threshold time period where the fairshare pricing obtains higher cumulative utility than any class of exploitative pricing is:

$$t^* := \sup_{p_t < p_t^*, \forall t} \left\{ T \in \mathbb{N} : \mathbb{E}\left[ \sum_{t=0}^{T} \delta^t \left( (u_t - p_t^*) - \prod_{i=0}^{t-1} \pi(p_i, p_i^*)(u_i - p_i) \right) \right] \leq 0 \right\}. \quad (11)$$

And $t^*$ is increasing as $\delta$ decreases.

We run detailed robustness check with different values of $\delta$ for our experiments in Appendix E.2.

## 4 Empirical Analyses: Benefits of Fairshare Pricing

This section evaluates the LLM data market and the proposed *fairshare* pricing framework.

### 4.1 Experimental Setup

In our experiments, a buyer is equipped with a single LLM, and each seller owns one data sample. Buyers seek to buy training data to improves task-specific model performance (e.g.,math problem solving, medical diagnosis, or physical reasoning), which in turn increases their *utility*. (We assume an affine mapping between performance and utility; see Appendix C.1.)

**Buyers and Models.** We consider three buyers, each using a standard open-source LLMs: Llama-3.2-Instruct-1b [81], Pythia-1b, and Pythia-410m [82]. These models are pre-trained on different corpora and exhibit varying preferences for downstream post-training data [83].

**Sellers and Datasets.** We focus on challenging, human-annotated tasks: MathQA and GSM8K [27, 84] for math, MedQA [28] for medical diagnosis, and PIQA [29] for physical reasoning [85–87]. Table 1 in Appendix F shows dataset splits and examples. We use the training splits as seller data and simulate the market dynamics from Section 3, treating each task as a market scenario.

### 4.2 LLM Data Market Experiments

We first evaluate our pricing framework in terms of buyer and seller welfare.

**Market Design:** Following the general setups in Section 4.1, we simulate a market with 2 buyers and 10 sellers over multiple time steps. To examine buyers with varying resources, we include a high-budget buyer (well-funded LLM builder) and a low-budget buyer (under-resourced one). Each buyer's budget is randomly drawn from distributions with different mean. At each time step, (1) sellers arrive sequentially with a new dataset ( 300 data samples) at a fixed price, and then (2) once all sellers arrive, buyers make purchases by solving Equation (2). Full details are in Appendix E.2.

**Participation Function:** Following Assumptions 1 to 2, we simulate 100 time steps with a discount factor $\delta = 0.98$. We also run experiments with different value of $\delta$ showing consistent and robust results. (See Appendix E). Seller participation probability is defined as $\pi(p_{j,t}, p_{j,t}^*) = p_{j,t}/p_{j,t}^*$ for its simplicity and compliance with Assumptions 1 and 1.1. Sellers receiving exploitative pricing ( $p_{j,t} < p_{j,t}^*$) are less likely to participate.

**Pricing Methods:** We consider four pricing methods: (1) *Fairshare* – $p_{j,t} = p_{j,t}^*$ (our *fairshare* pricing framework); (2) *Reduced* – a fixed discount of *fairshare* price, $p_{j,t} = c * p_{j,t}^*$ with $c = 0.5$; (3) *Random* – random drawn from $(0, p_{j,t}^*)$; (4) *Exploitative* – fixed low price (10% of the avg. utility).

Figure 4 compares the overall welfare outcomes of different pricing methods for buyers and sellers using Pythia-1b on the MedQA task. Results on MathQA and PIQA (with Pythia-410m and Llama-3.2-Instruct-1b) in Appendix F show similar patterns.

**Exploitative Pricing Leads to Lose-Lose Outcomes:** The exploitative pricing method, which sets uniformly low, fixed prices to reflect real-world practices ([11]), offers LLM developers short-term utility gains (see Figures 4a and 4b). However, this approach systematically undervalues datasets and unfairly compensates annotators. As a result, sellers exit the market over time, triggering a collapse in data supply. Even well-funded LLM developers are unable to source sufficient data – hindering the advancement of LLMs.

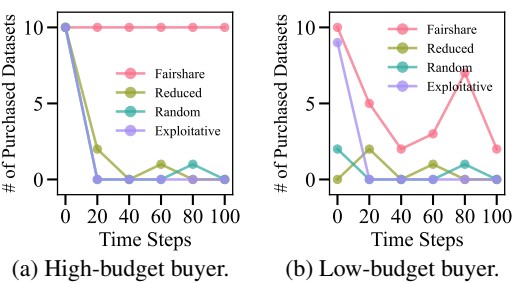

(a) High-budget buyer.  (b) Low-budget buyer.

Figure 5: Purchased datasets for the buyer with high budget (Figure 5a) and low budget (Figure 5b) over 100 time periods. Pythia-1b; MedQA.

**Fairshare Pricing Leads to Win-Win Outcomes:** For sellers, *fairshare* pricing consistently yields the highest profit over time (Figure 4c), aligning with predictions from our theoretical model (Section 3.1). For buyers, *fairshare* pricing framework is particularly effective for the high-budget buyer (Figure 4a), maximizing long-term utility. However, the low-budget buyer (Figure 4b) experiences reduced short-term utility in exchange for long-term gains. Its limited budget prevents them from fully leveraging the increased dataset supply ensured by *fairshare* pricing, making other low-pricing methods initially more appealing. Yet, in the long run, *fairshare* pricing sustains seller participation, ensuring data supply.

## 4.3   Ablation Study: Effect of Different Data Valuation Methods

This experiment assesses the impact of different *data valuation* strategies in the *fairshare* framework.

**Setup.** Using the models and datasets introduced earlier in Section 4.1, we run separate simulations for each market (math, medical, physical reasoning), testing different *data valuation* methods for the buyer. Each buyer is randomly assigned to use one of four valuation methods to select training data and fine-tune their model. The seller receives payments according to the data's assigned value. Valuation scores are normalized to $[0, 1]$ for pricing compatibility.

**Data Valuation Methods.** We consider the following methods, where each assigns a value to every data sample: (1) *Constant* baseline – assigns the same value, mimicking flat-rate pricing on platforms; (2) *Random* baseline – values drawn uniformly from $[0, 1]$; (3) *Semantic* – uses BM25 [30] to compute average similarity to the representative set; (4) *Influence-based:* returns a score which leverages learning gradients to estimate a data sample's avg. contribution to model learning of a representative dataset. Specifically, we use $\text{Infl}_{\text{IP}}$ [56, 22] and DataInf [21], which are influence-based methods adapted for the LLM realm. See Appendix A for additional details.

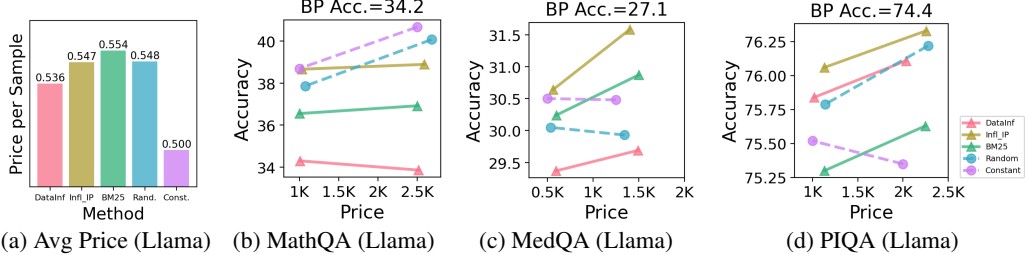

Figure 6: *Left column:* Average price-per-sample cost of purchased data across math, medical, and physical reasoning data markets using different data valuation methods. *Right, middle-columns:* Buyers' model performance versus cost, *before purchasing* (BP) data, and after purchasing 2K and 4K data samples. Additional analysis on the Pythia-1b/410m models are in Appendix F.

Furthermore, we adopt a one-step training approach for data valuation, where the value of each data sample is estimated by performing a single training step on it and measuring the resulting change in model performance relative to the original model. Results are presented in in Appendix F.2. This approach serves as an "oracle" for influence-based methods [23, 56], as it directly quantifies data value through model training, although it is more computationally expensive than the previous listed methods.

**Market/Pricing Setup:** We reserve $1\%$ of the samples from each dataset's training split to represent the existing data in their respective markets. Each data sample was randomly priced between $(0, 1]$. Next, for each remaining data sample in the training set, we determine whether each buyer will purchase the data sample at potential price points $[0.5, 0.625, 0.75, 0.875, 1.0]$ by solving Equation (2). The seller then sets their prices according to Equation (4). We price data separately for each data valuation method. This assesses the method's ability to discern whether a new data sample is worth purchasing for each buyer given the existing market data, as noted in our analysis in Section 3. Further details on all experiments are described in Appendix E.1.

**Results.** Figure 6 presents results across *data valuation* methods. Buyers using a valuation method (ie., BM25, $\text{Infl}_{\text{IP}}$, DataInf) in general achieved higher model performance across tasks. When considering the trade-off between cost and performance, $\text{Infl}_{\text{IP}}$ offered the best balance, delivering strong model improvements at a lower cost than constant, random, and BM25 (Figures 6a and 9a). Our results highlight the benefits of learning-aware data valuation methods. By prioritizing high-impact data, they offer a better alternative for buyers, particularly those with limited budgets.

We also run the error analysis of *data valuation* methods. Following previous studies [71, 72], we compute the correlation between Oracle and $\text{Infl}_{\text{IP}}$, reporting spearman correlation of $0.54, 0.42$ for MathQA and PIQA (see Figure 8 in Appendix F). This underscores the opportunity for future advances in data valuation accuracy and scalability, which can be seamlessly integrated into our flexible fairshare framework.

## 5  Conclusion

We introduced a *fairshare* pricing framework that leverages *data valuation* methods to enable transparent and equitable training data pricing for LLMs. Our results show that buyers achieve higher gains at lower cost. At the same time, sellers earn optimal prices for their data, fostering a win-win outcome that enhances long-term market sustainability and social welfare. To our knowledge, this is the first work to integrate game-theoretic pricing models with LLM-specific data valuation techniques, addressing the real-world dynamics of the emerging data market. Our framework offers actionable insights for policymakers and regulators aiming to ensure fairness and transparency in LLM training data markets. By fostering fair market access, our framework also empowers small businesses and startups, leading to more equitable technological advancements. Future work could extend our framework to incorporate additional *data valuation* methods, incomplete information settings (e.g., Bayesian games), and diverse data domains (e.g., pretraining vs. fine-tuning). We hope this work paves a fruitful way for future research in equitable markets for AI and emerging technologies.

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

# A  Influence-based Data Valuation

In Section 4, we introduced a gradient-based data attribution method, denoted as $\text{Infl}_{\text{IP}}$. In this section, we provide additional information on $\text{Infl}_{\text{IP}}$, which has been shown to be effective in training data selection in previous works [56, 22]. Suppose we have a LLM parameterized by $\theta$, and a train set $D$ and a test set $\mathcal{D}'$. For a training sample $d \in D$, we wish to estimate its training impact on a test sample $d' \in D$. That is, we want to measure the impact of $d$ on the model's loss on $d'$ (i.e., $\mathcal{L}(d'; \theta)$). As simple method of achieving this is to take training step – that is, a gradient descent step – on $d$ and obtain:

$$\hat{\theta} = \theta - \eta \nabla \mathcal{L}(d; \theta) \tag{12}$$

where $\eta$ is the learning rate. Then, in order to measure the influence of $d$ towards $d'$, we wish to find the change in loss on $d'$:

$$\mathcal{L}(d'; \theta) - \mathcal{L}(d'; \hat{\theta}) \tag{13}$$

Instead of taking a single training step to measure the influence of $d \in D$ on $d'$, we can instead approximate Equation (13) with using the following:

**Lemma 5.** Suppose we have a LLM with parameters $\theta$. We perform a gradient descent step with training sample $d$ with learning rate $\eta$ such that $\hat{\theta} = \theta - \eta \nabla \mathcal{L}(d; \theta)$. Then,

$$\mathcal{L}(d'; \theta) - \mathcal{L}(d'; \hat{\theta}) \approx \nabla \mathcal{L}(d'; \theta) \cdot \nabla \mathcal{L}(d; \theta)$$

See Appendix D for the proof.

Then, we set $\text{Infl}_{\text{IP}}$ to be:

$$\text{Infl}_{\text{IP}} = \nabla \mathcal{L}(d'; \theta) \cdot \nabla \mathcal{L}(d; \theta) \tag{14}$$

which is the dot-product between the learning gradients of $d'$ and $d$.

# B  Royalty model

So far, we have shown the case of *flat rate* (see Section 3.1), which is well-suited resource-rich buyers, such as leading tech companies whose LLMs generate significant economic value due to their wide-ranging impact and scalability. In this section, we introduce the *royalty model*, a contract framework that differs from the flat rate by offering a subscription-like structure. Under the royalty model, the price paid for training data is proportional to the future economic value generated by the LLM, providing a flexible and performance-based approach to data valuation. This scenario incorporates buyers in a less dominant position – those who are (1) uncertain about the prospective model outcome or (2) do not own a sufficient cash flow for purchasing data with full prices. We present updated decision-making models for buyers and sellers as follows.

**Buyers.** Unlike the flat pricing setting, the buyer $B_k$ would alternatively pay with a *fractional price*. Suppose each dataset $D_j$ is priced with an individual rate $\alpha_j \in [0, 1)$ (as we denote $\boldsymbol{\alpha} = (\alpha_1, \cdots, \alpha_N)$), then the price of an arbitrary data collection $u_k(\boldsymbol{x})$ is a fraction of its future marginal gain, i.e., $\boldsymbol{x}^T \boldsymbol{p} = f(\boldsymbol{\alpha}, \boldsymbol{x}) u_k(\boldsymbol{x})$, where the overall rate function $f : [0, 1)^{|\boldsymbol{\alpha}|} \times \{0, 1\}^{|\boldsymbol{x}|} \to [0, 1)$ depends on specific contexts. We assume that $f$ is a monotonically non-decreasing function of $\boldsymbol{\alpha}$. In this sense, the buyer $B_k$ reduces the risk of losing $b_k$ from its cash flow while the seller is betting on the potential value of the LLM $M_k$. Then we obtain an updated objective function for $B_k$:

$$g_{k,N,\text{frac}}(\boldsymbol{x}) := (1 - f(\boldsymbol{\alpha}, \boldsymbol{x})) u_k(\boldsymbol{x}), \tag{15}$$

On the other hand, similar to the budget constraint (see Equation (2)), here each buyer $B_k$ has a maximum rate $\overline{\alpha}_k$ that it is willing to pay. Then the buyer's purchasing problem is given as

$$\tilde{\boldsymbol{x}}^{k,N,\text{frac}} := \underset{\boldsymbol{x} \in \mathcal{X}_{k,N,\text{frac}}}{\arg\max} \ g_{k,N,\text{frac}}(\boldsymbol{x}), \quad \text{s.t.} \tag{16}$$

$$\mathcal{X}_{k,N,\text{frac}} := \{\boldsymbol{x} \mid g_{k,N,\text{frac}}(\boldsymbol{x}) \geq 0, f(\boldsymbol{\alpha}, \boldsymbol{x}) \leq \overline{\alpha}_k\}, \tag{17}$$

And $\tilde{\boldsymbol{x}}^{k,N,\text{frac}}$ is the optimal solution to $\max_{\boldsymbol{x} \in \mathcal{X}_{k,N,\text{frac}}} g_{k,N,\text{frac}}(\boldsymbol{x})$ with a given rate vector $\boldsymbol{\alpha}$.

**Sellers.** In the fractional pricing setting, since the buyer $B_k$ pays for the entire data collection, there should exists a fair and transparent allocation mechanism that distributes a portion of the total price charged to each individual dataset $D_j$. That is, $\boldsymbol{x}_j p_j = \sum_{k=1}^{M} f_j(\boldsymbol{\alpha}, \tilde{\boldsymbol{x}}^{k,N,\text{frac}}) u_k(\tilde{\boldsymbol{x}}^{k,N,\text{frac}})$, where $f(\cdot) = \sum_{j=1}^{N} f_j(\cdot)$. And we assume that for all $j \in [N]$, $f_j(\cdot)$ is monotonically non-decreasing over $\boldsymbol{\alpha}$. Therefore, we have an updated profit function for $Se_j$:

$$r_{\text{frac}}(\alpha_j) := \sum_{k=1}^{M} f_j(\boldsymbol{\alpha}, \tilde{\boldsymbol{x}}^{k,N,\text{frac}}) u_k(\tilde{\boldsymbol{x}}^{k,N,\text{frac}}) - c_j. \tag{18}$$

which gives the following problem:

$$\alpha_j^* := \underset{\alpha_j \in \mathcal{A}_{j,M,\text{frac}}}{\arg\max} \; r_{\text{frac}}(\alpha_j), \quad \text{s.t.} \tag{19}$$

$$\mathcal{A}_{j,M} := \{\alpha_j \in [0,1) \mid r_{\text{frac}}(\alpha_j) \geq 0\}, \tag{20}$$

From this point onward, the market dynamics stays the same as in the previous section. It is noted that, compared to $\max_{p_j \in \mathcal{P}_{j,M}} r(p_j)$ where optimal flat rate is indirectly connected to the *utility*, the optimal rate of $\max_{\alpha_j \in \mathcal{A}_{j,M,\text{frac}}} r_{\text{frac}}(\alpha_j)$ offers a more direct representation of the *utility*.

## B.1 Solving for optimal price of the royalty model

Similar to *flat rate*, in the case of *royalty model*, we need to solve buyer's problem $\max_{\boldsymbol{x} \in \mathcal{X}_{k,N-1,\text{frac}}} g_{k,N-1,\text{frac}}(\boldsymbol{x})$ (before the arrival of $S_j$) and $\max_{\boldsymbol{x} \in \mathcal{X}_{k,N,\text{frac}}} g_{k,N,\text{frac}}(\boldsymbol{x})$ (after the arrival of $S_j$) for all $k \in [M]$ and seller's problem $\max_{\alpha_j \in \mathcal{A}_{j,M,\text{frac}}} r_{\text{frac}}(\alpha_j)$.

**Solve buyer's problems.** For each feasible collection of datasets $\boldsymbol{x} \in \mathcal{X}_{k,N-1,\text{frac}}$ (before the arrival of $S_j$), denotes it union with dataset $D_j$ as $\boldsymbol{x}^{\text{new}}$. Then we run the check: (1) if the *net utility* of $\boldsymbol{x}^{\text{new}}$ is larger than the one of $\tilde{\boldsymbol{x}}^{k,N-1,\text{frac}}$, i.e., $g_{k,N,\text{frac}}(\boldsymbol{x}^{\text{new}}) > g_{k,N,\text{frac}}(\tilde{\boldsymbol{x}}^{k,N-1,\text{frac}})$, and (2) if the rate for purchasing $\boldsymbol{x}^{\text{new}}$ is still under the budget $\overline{\alpha}_k$, i.e., $f([\boldsymbol{\alpha}^T \; \alpha_j], \boldsymbol{x}^{\text{new}}) \leq \overline{\alpha}_k$, where $[\boldsymbol{\alpha}^T \; \alpha_j]$ denotes concatenating $\alpha_j$ to $\boldsymbol{\alpha}$. If the answer is positive to bother tests, then we can determine that the buyer $B_k$ will change its decision and purchase $S_j$ under the rate $\alpha_j$.

**Solve seller's problem.** First, we consider when $\alpha_j = 0$. We could first find all $\boldsymbol{x} \in \mathcal{X}_{k,N-1,\text{frac}}$ such that $g_{k,N,\text{frac}}(\boldsymbol{x}^{\text{new}}) > g_{k,N,\text{frac}}(\tilde{\boldsymbol{x}}^{k,N-1,\text{frac}})$. And we denote the set that contains such $\boldsymbol{x}$ as $\mathcal{X}_{k,N-1,\text{frac}}^1$. If $\mathcal{X}_{k,N-1,\text{frac}}^1$ is empty, then $\mathbb{1}_{\{B_k, D_j, p_j\}} = 0$ (indicating whether $B_k$ will purchase $D_j$ at price $p_j$ or not), as $S_j$ cannot bring positive value to $B_k$; else, then for all $\boldsymbol{x} \in \mathcal{X}_{k,N-1,\text{frac}}^1$, thanks to the monotonicity of $f_j$ over $\alpha_j$, we could gradually increase $\alpha_j$ until the either of the two criterion are met first: (1) we find the largest $\alpha_j$ such that $g_{k,N,\text{frac}}(\boldsymbol{x}^{\text{new}}) > g_{k,N,\text{frac}}(\tilde{\boldsymbol{x}}^{k,N-1,\text{frac}})$, and (2) $f_j([\boldsymbol{\alpha}^T \; \alpha_j], \boldsymbol{x}^{\text{new}}) \leq \overline{\alpha}_k$. Then we have the following property about the optimal rate $\alpha_j^*$ for Equation (19):

**Lemma 6** (Characterization of $\alpha_j^*$ under *royalty model*). Define $\alpha_j^{\boldsymbol{x}}$ as

$$\min \left\{ \sup_{\alpha_j \in [0,1)} \left\{ \alpha_j : f_j([\boldsymbol{\alpha}^T \; \alpha_j], \boldsymbol{x}^{\text{new}}) < 1 - (1 - f_j(\boldsymbol{\alpha}, \tilde{\boldsymbol{x}}^{k,N-1,\text{frac}})) \frac{u_k(\tilde{\boldsymbol{x}}^{k,N-1,\text{frac}})}{u_k(\boldsymbol{x}^{\text{new}})} \right\}, \right.$$

$$\left. \sup_{\alpha_j \in [0,1)} \left\{ \alpha_j : f_j([\boldsymbol{\alpha}^T \; \alpha_j], \boldsymbol{x}^{\text{new}}) \leq \overline{\alpha}_k \right\} \right\}. \tag{21}$$

For every $\boldsymbol{x} \in \mathcal{X}_{k,N-1,\text{frac}}^1$ and all $k \in [M]$, we obtain $\alpha_j^{\boldsymbol{x}}$ and their union $\cup_{k=1}^{M} \cup_{\boldsymbol{x} \in \mathcal{X}_{k,N-1,\text{frac}}^1} \{\alpha_j^{\boldsymbol{x}}\}$. Then we have $\alpha_j^* \in \cup_{k=1}^{M} \cup_{\boldsymbol{x} \in \mathcal{X}_{k,N-1,\text{frac}}^1} \{\alpha_j^{\boldsymbol{x}}\}$.

*Remark* B.1 (Similarities between *flat rate* and *royalty model*). Observing from Lemmas 2 and 6, we see that the both the optimal price $p_j^*$ and the optimal rate $\alpha_j^*$ are closely tied to $B_k$'s *maximum willingness to pay*. That is, compared to the market prior to the arrival of $S_j$, the optimal values are characterized by the minimum of two factors: (1) *marginal utility* that $S_j$ provides to $B_k$ and (2) $B_k$'s *budget surplus*. It is also noted that, under *royalty model*, the rate function $f$ also plays an important role as it determines the how the single rate $\alpha_j$ affects the total rate that $B_k$ pays.

# C Applications for Real-Life Scenarios

In real-life settings, the relationship between the data valuation of a training sample and the buyer's utility $u_k$ (i.e., the economical value, which may be expressed in dollar amounts) can have different mappings, as mentioned in Section 3.1. Suppose the data valuation function is denoted as $v_k : D \to \mathbb{R}$ for a dataset $D$. Then, a buyer may expect a linear relationship between $v_k$ and $u_k$, where the utility increases as the data valuation score increases. Alternatively, a buyer may prefer to only purchase data beyond a certain threshold for $v_k$. In this section, we present three types mappings between $v_k$ and $u_k$ to reflect these scenarios: *linear*, *discrete*, and *zero-one* mappings. We show that these mappings can be easily adapted to our proposed framework in Section 3. We only present the updated buyer's purchasing problem (Equation (2)) since the seller's pricing problem (Equation (4)) stays the same.

## C.1 Linear Outcome

In practice, there are many applications where $u_k$ is an affine function of $v_k$. As previously mentioned, training LLMs on data with higher valuation scores $v_k$ can result in better economic value towards downstream model performance, as shown in previous works [22, 23]. In this outcome setting, in addition to considering $u_k$ to be a affine function of $v_k$, we also include a bias variable $\beta$ to account for other potential other factors that are independent of $v_k$. Therefore, we can set $u_k = \gamma v_k(\boldsymbol{x}) + \beta$ into Equation (1), where $\gamma \in \mathbb{R}_+$ is a known coefficient, and obtain buyer $B_k$'s net utility function for the linear outcome:

$$g_{k,N}(\boldsymbol{x}) = \gamma v_k(\boldsymbol{x}) + \beta - \boldsymbol{x}^T \boldsymbol{p}, \tag{22}$$

To obtain optimal price $p^*$, we can directly refer to same procedure described in Section 3 using set values for $\gamma$ and $\beta$.

## C.2 Discrete Outcome

There are also many applications where $u_k$ is discrete. For instance, if the data buyers are participating in an LLM benchmark challenge, such as MMLU [88], then training on data that falls within various ranges $v_k$ may lead to drastically different model performance, and hence leaderboard rankings.

To mirror this, consider $u_k$ to be a category variable. We denote $\{c_h\}_{h=1}^{H}$ as a strictly increasing set of numbers such that when $v_k \in [c_h, c_{h+1})$, the buyer will receive reward $u_{k,h}$. We also assume that $u_{k,h+1} > u_{k,h}$ since higher data valuation scores may lead to a larger reward. Therefore, we could set $u_k = \sum_{h=1}^{H} \mathbb{1}_{\{v_k(\boldsymbol{x}) \in [c_h, c_{h+1})\}} u_{k,h}(\boldsymbol{x})$ and rewrite buyer $B_k$'s net utility function as

$$g_{k,N}(\boldsymbol{x}) = \sum_{h=1}^{H} \mathbb{1}_{\{v_k(\boldsymbol{x}) \in [c_h, c_{h+1})\}} u_{k,h}(\boldsymbol{x}) - \boldsymbol{x}^T \boldsymbol{p}. \tag{23}$$

We again apply the same procedure in Section 3 to solve for the optimal pricing.

## C.3 Zero-One Outcome

There are scenarios where the data buyers are risk-adverse and focus on the effects of rare events. In these cases, suppose that $v_k$ is normalized between [0, 1]. Then buyers may wish to purchase training data with higher values of $v_k$, assuming that purchasing data with lower $v_k$ may result in severe adverse effects. For instance, data buyers who are building AI for healthcare should not purchase data with incorrect medical information, and even a small amount of contaminated data can result in severe real-life consequences such as mis-diagnosis [28, 89] or unsuitable medical protocols in emergency situations [90]. Therefore, in this context, we consider $u_k$ as a generalized Bernoulli distribution. The downstream outcome has a small positive reward $\underline{u}$ with probability $v_k$ (normal events) and a massive negative reward $\overline{u}$ with probability $1 - v_k$ (undesirable rare events). And we assume that $\mathbb{E}(u_k) > 0$. Therefore, we can plug in and obtain buyer $B_k$'s net utility function:

$$g_{k,N}(\boldsymbol{x}) = \mathbb{E}[u_k(\boldsymbol{x})] - \boldsymbol{x}^T \boldsymbol{p} \tag{24}$$

$$= v_k(\boldsymbol{x})(\underline{u} - \overline{u}) + \overline{u} - \boldsymbol{x}^T \boldsymbol{p}, \tag{25}$$

which is an affine function of $v_k$. Therefore, we again apply same procedure in Section 3 to solve for the optimal pricing.

## C.4 Multiple tasks

In practice, many LLMs are evaluated over multiple tasks [88]. To this end, we consider the context where buyer $B_k$ wishes their model $\mathcal{M}_k$ to perform well across multiple tasks, denoted as $Q$. Each data valuation score for a task is denoted by $v_1^k, \cdots, v_Q^k$ and the vector of all task valuations is denoted as $\boldsymbol{v_k} = (v_{k,1} \cdots v_{k,Q})$. Then we consider that the utility $u_k$ is an affine function of the utility in each task, denoted by $\boldsymbol{u_k} = (u_{k,1} \cdots u_{k,Q})$ that is, $u_k = \boldsymbol{\theta}^T \boldsymbol{u_k} + \epsilon$, where $\boldsymbol{\theta} \in \mathbb{R}^Q$ is a coefficient vector and $\epsilon \in \mathbb{R}$ denotes other factors independent from $\boldsymbol{u_k}$. We also assume that the each task is one of three categories mentioned in the last section. Therefore, we can rewrite $\boldsymbol{u_k}$ as a function of $\boldsymbol{v_k}$, which gives $u_k = \boldsymbol{\theta}^T \boldsymbol{u_k}(\boldsymbol{v_k}) + \epsilon$. Therefore, the buyer's net utility function becomes

$$g_{k,N}(\boldsymbol{x}) = \boldsymbol{\theta}^T \boldsymbol{u_k}(\boldsymbol{v_k}(\boldsymbol{x})) + \epsilon - \boldsymbol{x}^T \boldsymbol{p}. \tag{26}$$

whose solution could adopt the same procedure as described in Section 3 to solve for the optimal pricing.

# D  Lemmas and Proofs

**Lemma 1.** With Assumptions 1 to 2, any exploitative pricing (i.e., $p_t < p_t^*, \forall t$) will only maximize cumulative utility within a finite horizon – after which it is strictly suboptimal.

*Proof.* Lemma 1 equivalently states that with Assumptions 1 to 2, the optimal pricing strategy for the buyer is also the ideal price $p_t^*$. Thus, we show that when the buyer pays the ideal price $p_t^*$, its total value is the largest, i.e.,

$$\mathbb{E}\left[u_t - p_t^* + \delta \mathbb{E}\left[r(p_t^*, p_t^*) G \mid u_t, b_t\right]\right] \geq \mathbb{E}\left[u_t - p_t + \delta \mathbb{E}\left[\pi(p_t, p_t^*) G \mid u_t, b_t\right]\right] \tag{27}$$

for all $p_t \in [0, p_t^*)$.

The seller will not set a price above its ideal price $p_t^*$ as it will decreases its profit, since the ideal price $p_t^*$ should be its profit-maximizing price. Any $p_t > p_t^*$ results lower profits.

Also, the buyer will not accept a price $p_t > p_t^*$, since it decreases the net *utility* gain $u_t - p_t$ without increasing seller's participation probability $\prod_{t=0}^{T-1} \pi(p_t, p_t^*)$.

Through some linear transformation, this is equivalent to show that

$$\mathbb{E}\left[p_t - p_t^* + \delta \mathbb{E}\left[G(\pi(p_t^*, p_t^*) - \pi(p_t, p_t^*)) \mid u_t, b_t\right]\right] \geq 0. \tag{28}$$

We first find the lower bound of $G$. We see that $p_t^*$ gives a payoff of

$$\mathbb{E}\left[\sum_{t=0}^{\infty} \delta^t (u_t - p_t^*)\right] \geq \frac{\min_{t \in [0,\infty)} \mathbb{E}[u_t - p_t^*]}{1 - \delta}. \tag{29}$$

Therefore, we must have $G \geq \frac{\min_{t \in [0,\infty)} \mathbb{E}[u_t - p_t^*]}{1-\delta}$. Along with Assumptions 1 to 2, this gives us, for a given $u_t$ and $b_t$,

$$\frac{\delta G\left(\pi(p_t^*, p_t^*) - \pi(p_t, p_t^*)\right)}{p_t^* - p_t} \geq \delta G L \geq 1, \tag{30}$$

implying that

$$\mathbb{E}\left[p_t - p_t^* + \delta \mathbb{E}\left[G(\pi(p_t^*, p_t^*) - \pi(p_t, p_t^*)) \mid u_t, b_t\right]\right] \geq 0. \tag{31}$$

$\square$

**Lemma 7** (*Participation Loss Is Lower-Bounded*)**.** With Assumption 1, let $P := \lim_{T \to \infty} P_T$ and $S := \sum_{i=0}^{\infty} (1 - \pi(p_i, p_i^*))$. Then for the class of all exploitative pricing strategies where $p_t \leq p_t^*, \forall t$, we have (1) $P_T$ is strictly decreasing and (2) The limit of reduced participation is lower bounded, i.e., $1 - P \geq 1 - e^{-S} > 0$.

*Proof.* The first part is trivial to see as we assume that $\pi(p_i, p_i^*) < 1$ for all $p_i < p_i^*$ from Assumption 1.

For the second part, we first denote $P := \lim_{t \to \infty} P_t$. Then we take the following transformation:

$$\log P = \log \left( \prod_{i=0}^{\infty} \pi(p_i, p_i^*) \right) = \sum_{i=0}^{\infty} \log \left( \pi(p_i, p_i^*) \right). \tag{32}$$

Since we have $0 < \pi(p_i, p_i^*) < 1$, then $\log \left( \pi(p_i, p_i^*) \right) \le \pi(p_i, p_i^*) - 1$, indicating that

$$\log P = \sum_{i=0}^{\infty} \log \left( \pi(p_i, p_i^*) \right) \le \sum_{i=0}^{\infty} \left( \pi(p_i, p_i^*) - 1 \right) = -S, \tag{33}$$

which gives $P \le e^{-S}$ by exponentiating both sides. $\qquad\square$

**Lemma 2.** Seller $S_j$'s optimal price for $D_j$ is characterized as one of buyers' MWP:

$$p_j^* \in \cup_{k=1}^{M} \text{MWP}_k. \tag{34}$$

*Proof.* Recall that without dataset $D_j$, each buyer $B_k$ has already solved $\max_{\boldsymbol{x} \in \mathcal{X}_{k,N-1}} g_{k,N-1}(\boldsymbol{x})$ according to our market definition in Section 3, where $\mathcal{X}_{k,N-1}$ is the set of all feasible purchase decisions. Next, after seller $S_j$ (with $D_j$) has arrived on the market, we analyze the conditions in which $B_k$ will purchase $D_j$ at a potential price $p_j$. For each feasible purchase decision (i.e., a collection of datasets), represented by $\boldsymbol{x} \in \mathcal{X}_{k,N-1}$, let $\boldsymbol{x} + e_j$ denote its union with $D_j$, where $e_j$ is the unit vector indicating $D_j$ is selected. For buyer $B_k$ to change their previous decision to purchase $D_j$, there are two requirements that need to be satisfied. First, we must have:

$$g_{k,N}(\boldsymbol{x} + e_j) > g_{k,N-1}(\tilde{\boldsymbol{x}}^{k,N-1}). \tag{35}$$

That is, the *net utility* $g_{k,N}(\boldsymbol{x} + e_j)$ of purchasing decisions $\boldsymbol{x} + e_j$, must be larger than the *net utility* $g_{k,N-1}(\tilde{\boldsymbol{x}}^{k,N-1})$ of a previous optimal purchasing decision $\tilde{\boldsymbol{x}}^{k,N-1}$. It is also noted that $g_{k,N}(\tilde{\boldsymbol{x}}^{k,N-1}) = g_{k,N-1}(\tilde{\boldsymbol{x}}^{k,N-1})$. Second, for buyer $B_k$ to purchase $\boldsymbol{x} + e_j$ at price $p_j$, we must fulfill the budget constraint:

$$p_j \le b_k - \left( \tilde{\boldsymbol{x}}^{k,N-1} \right)^T \boldsymbol{p} = \Delta b_k(\tilde{\boldsymbol{x}}^{k,N-1}), \tag{36}$$

which ensures that purchasing $D_j$ does not exceed the buyer's budget $b_k$. If both requirements are satisfied, then the buyer $B_k$ will change their previous purchasing decision in order to purchase $D_j$ under the price $p_j$. This procedure is presented in detail in Algorithm 1 in Appendix F.1.

Next, given the conditions for the buyer $B_k$ to purchase $D_j$, the seller must solve $\max_{p_j \in \mathcal{P}_{j,M}} r(p_j)$ to find the optimal price $p_j^*$. First, we consider an edge case where the price of dataset $D_j$ is set as $p_j = 0$. For a buyer $B_k$, we denote $\mathcal{X}_{k,N}^1$ as the set of all purchasing decisions where including $D_j$ in the purchase improves the buyer's previous net utility $g_{k,N}(\tilde{\boldsymbol{x}}^{k,N-1})$. That is, for every $\boldsymbol{x} + e_j \in \mathcal{X}_{k,N}^1$, we have $g_{k,N}(\boldsymbol{x} + e_j) > g_{k,N}(\tilde{\boldsymbol{x}}^{k,N-1})$. If $\mathcal{X}_{k,N}^1$ is empty, then $B_k$ will not purchase $D_j$ at any price, since $D_j$ cannot bring positive improved *net utility* to $B_k$. Then, when $p_j$ gradually increases and exceeds $\max_{\boldsymbol{x} \in \mathcal{X}_{k,N-1}} \{\min\{\Delta u_k(\boldsymbol{x} + e_j), \Delta b_k(\tilde{\boldsymbol{x}}^{k,N-1})\}\}$, then $B_k$ will decide not to purchase $D_j$, causing the value of $\sum_{k=1}^{M} \tilde{\boldsymbol{x}}_j^{k,N}(\boldsymbol{p})$ to drop by one. Since the profit function $r(p_j)$ is a piecewise linear function, the its optimal point must be one of its breakpoints. $\quad\square$

**Lemma 3** The optimal price for the seller $S$ under our framework is

$$p_t^* = \min\{u_t, b_t\}, \forall t. \tag{37}$$

With assumptions 1 to 2, $p_t^*$ gives the buyer the maximum cumulative *net utility* over infinite horizon.

*Proof.* In a single buyer and seller setting, we could trivially see that the optimal price for the seller $p_t^* = \min\{u_t, b_t\}$: if $p_t > p_t^*$, then the buyer would not purchase this dataset since the *net utility* would be negative; if $p_t < p_t^*$, then the seller's profit is not maximized.

Further, we refer to the proof of Lemma 1 for the second part. $\quad\square$

**Lemma 4** The threshold time period where the fairshare pricing obtains higher cumulative utility than any class of exploitative pricing is:

$$t^* := \sup_{p_t < p_t^*, \forall t} \left\{ T \in \mathbb{N} : \mathbb{E}\left[ \sum_{t=0}^{T} \delta^t \left( (u_t - p_t^*) - \prod_{i=0}^{t-1} \pi(p_i, p_i^*)(u_i - p_i) \right) \right] \le 0 \right\}. \tag{38}$$

And $t^*$ is increasing as $\delta$ increases.

*Proof.* For any given class of exploitative pricing strategy, when $\delta$ increases, the part: $\sum_{t=0}^{T} \delta^t \left( (u_t - p_t^*) - \prod_{i=0}^{t-1} \pi(p_i, p_i^*)(u_i - p_i) \right)$ increases. Therefore, for all class of exploitative pricing strategy, i.e., $p_t < p_t^*, \forall t$, then $t^*$ also decreases. $\qquad \square$

**Lemma 5.** Suppose we have a LLM with parameters $\theta$. We perform a gradient descent step with training sample $d$ with learning rate $\eta$ such that $\hat{\theta} = \theta - \eta \nabla \mathcal{L}(d; \theta)$. Then,

$$\mathcal{L}(d'; \theta) - \mathcal{L}(d'; \hat{\theta}) \approx \nabla \mathcal{L}(d'; \theta) \cdot \nabla \mathcal{L}(d; \theta)$$

*Proof.* First, we consider the change in loss of $z'$ using a first-order approximation:

$$\mathcal{L}(d'; \hat{\theta}) = \mathcal{L}(d'; \theta) + \nabla \mathcal{L}(d'; \theta)(\hat{\theta} - \theta) + \mathcal{O}(||\hat{\theta} - \theta||^2) \tag{39}$$

$$\mathcal{L}(d'; \theta) - \mathcal{L}(d'; \hat{\theta}) = -\nabla \mathcal{L}(d'; \theta)(\hat{\theta} - \theta) + \mathcal{O}(||\hat{\theta} - \theta||^2) \tag{40}$$

Next, suppose a gradient descent step is taken on training sample $d$, and the model parameters are updated as: $\hat{\theta} = \theta - \eta \nabla \mathcal{L}(d; \theta)$. Thus, we have $\hat{\theta} - \theta = -\eta \nabla \mathcal{L}(d; \theta)$, and the change in loss can be written as

$$\mathcal{L}(d'; \theta) - \mathcal{L}(d'; \hat{\theta}) \approx \eta \nabla \mathcal{L}(d'; \theta) \cdot \nabla \mathcal{L}(z; \theta) \propto \nabla \mathcal{L}(d'; \theta) \cdot \nabla \mathcal{L}(d; \theta) \tag{41}$$

Given that $\eta$ is a constant. $\qquad \square$

**Lemma 6** Define $\alpha_j^{\boldsymbol{x}}$ as

$$\min \left\{ \sup_{\alpha_j \in [0,1)} \left\{ \alpha_j : f_j([\boldsymbol{\alpha}^T \ \alpha_j], \boldsymbol{x}^{\text{new}}) < 1 - (1 - f_j(\boldsymbol{\alpha}, \tilde{\boldsymbol{x}}^{k,N-1,\text{frac}})) \frac{u_k(\tilde{\boldsymbol{x}}^{k,N-1,\text{frac}})}{u_k(\boldsymbol{x}^{\text{new}})} \right\}, \right.$$
$$\left. \sup_{\alpha_j \in [0,1)} \left\{ \alpha_j : f_j([\boldsymbol{\alpha}^T \ \alpha_j], \boldsymbol{x}^{\text{new}}) \le \overline{\alpha}_k \right\} \right\}. \tag{42}$$

For every $\boldsymbol{x} \in \mathcal{X}_{k,N-1,\text{frac}}^1$ and all $k \in [M]$, we obtain $\alpha_j^{\boldsymbol{x}}$ and their union $\cup_{k=1}^{M} \cup_{\boldsymbol{x} \in \mathcal{X}_{k,N-1,\text{frac}}^1} \{\alpha_j^{\boldsymbol{x}}\}$. Then we have $\alpha_j^* \in \cup_{k=1}^{M} \cup_{\boldsymbol{x} \in \mathcal{X}_{k,N-1,\text{frac}}^1} \{\alpha_j^{\boldsymbol{x}}\}$.

*Proof.* We show that, for every $\boldsymbol{x} \in \mathcal{X}_{k,N-1,\text{frac}}^1$, $\alpha_j^{\boldsymbol{x}}$ gives the largest revenue of $\boldsymbol{x} + e_j$ for $S_j$ (note that $\boldsymbol{x} + e_j = \boldsymbol{x}^{\text{new}}$). Recall that in the main text, we need to increase $\alpha_j$ from zero until we find the largest $\alpha_j$ such that either of:

1. $g_{k,N,\text{frac}}(\boldsymbol{x} + e_j) > g_{k,N,\text{frac}}(\tilde{\boldsymbol{x}}^{k,N-1,\text{frac}})$,

2. $f_j([\boldsymbol{\alpha}^T \ \alpha_j], \boldsymbol{x} + e_j) = \overline{\alpha}_k$.

If we rewrite the first condition, we are essentially looking for $\alpha_j$ such that

$$\sup_{\alpha_j \in [0,1)} \left\{ \alpha_j : f_j([\boldsymbol{\alpha}^T \ \alpha_j], \boldsymbol{x} + e_j) < 1 - (1 - f_j(\boldsymbol{\alpha}, \tilde{\boldsymbol{x}}^{k,N-1,\text{frac}})) \frac{u_k(\tilde{\boldsymbol{x}}^{k,N-1,\text{frac}})}{u_k(\boldsymbol{x} + e_j)} \right\} \tag{43}$$

Then we see that the revenue that for each $\boldsymbol{x} \in \mathcal{X}_{k,N-1,\text{frac}}^1$, seller $S_j$ can make from buyer $B_k$ is

$$f_j([\boldsymbol{\alpha}^T \ \alpha_j], \boldsymbol{x} + e_j) u_k(\boldsymbol{x}_j(\boldsymbol{\alpha})), \tag{44}$$

where $f_j([\boldsymbol{\alpha}^T \ \alpha_j], \boldsymbol{x} + e_j)$ is a non-decreasing function over $\alpha_j$ while other terms stays fixed. It indicates that $\alpha_j^{\boldsymbol{x}}$ is the largest $\alpha_j$ that the seller $S_j$ could set for buyer $B_k$ to purchase $S_j$. Therefore, the optimal rate $\alpha_j^*$ is one of the rates $\cup_{k=1}^{M} \cup_{\boldsymbol{x} \in \mathcal{X}_{k,N-1,\text{frac}}^1} \{\alpha_j^{\boldsymbol{x}}\}$. $\qquad \square$

# E  Additional Experimental Details

## E.1  Data Valuation Experiments

For each dataset, we randomly sample 200 demonstrations from the validation set to form a representative dataset [4]. Each data sample in the market is then scored based on its similarity to this representative set.

**Model Training:** After obtaining purchasing decisions for all data samples, the buyers train their models using the purchased data. In order to conduct a fair comparison across buyers, we sample a set number of data from the buyers' purchases (shown in Figure 6). We train each model (i.e., buyer) on these samples separately using LoRA [91] for 3 epochs, with a learning rate of 2e-5 and batch size 32. All models are trained on A6000 GPUs on single GPU settings and take less than 1 hour.

**Model Evaluation:** For evaluation, we use the test splits of the previously mentioned datasets. In particular, we use 5-shot evaluation on the MathQA test set, and 4-shot evaluation in on the MedQA test. Table 2 in Appendix F shows the demonstrations used for 5-shot and 4-shot evaluation.

## E.2  Data Pricing Experiments

**Experiment Setups**: We simulate two buyer budgets at each time step $t$. The first buyer (high budget) has a budget uniformly randomly generated between $95\%$ and $100\%$ of the total utilities of all 10 datasets listed in the market. The second buyer (low budget) has a budget uniformly randomly generated between $90\%$ and $95\%$ of the total utilities of all 10 datasets listed in the market. At each time period, seller's arriving orders are randomly shuffled. And they prices their own datasets sequentially.

**Robustness Check.** As discussed in Lemma 4, the threshold $t^*$ when *fairshare* becomes optimal for the buyer) increases as the discount factor $\delta$ decreases. To run a robustness check, for the high-budget buyer in Figure 4a, setting $\delta = 0.999, 0.99, 0.98$ yields $t^* = 32, 38, 44$ respectively, which is consistent with our theoretical analysis. In below, we compare the change of buyer's accumulative utilities over different values of $\delta$.

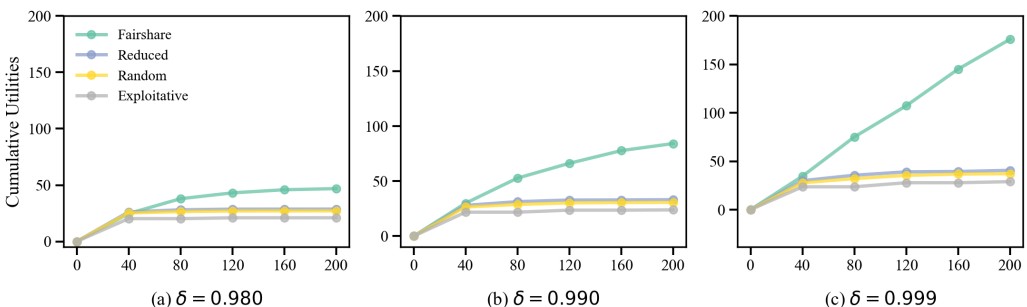

Figure 7: Buyer's accumulative utilities over time periods for $\delta = 0.980, 0.990, 0.999$.

---

[4]Note: For PIQA we take 200 samples from the training set since the validation set is commonly reserved for testing.

# F Additional Experiments and Figures

## F.1 Algorithms

---

**Algorithm 1** Determine if buyer $B_k$ will purchase dataset $D_j$ at price $p_j$

---

1: **Inputs:** prices $\boldsymbol{p}$, previous optimal $\tilde{\boldsymbol{x}}^{k,N-1}$, feasible set $\mathcal{X}_{k,N-1}$, price $p_j$, budget $b_k$
2: **Output:** indicator $\mathbb{1}_{\{B_k,D_j,p_j\}}$
3: Initialize $\mathbb{1}_{\{B_k,D_j,p_j\}} \leftarrow 0$
4: $p_j \leftarrow 0$
5: **for** $\boldsymbol{x} \in \mathcal{X}_{k,N-1}$ **do**
6: $\quad \boldsymbol{x}^{\text{new}} \leftarrow \boldsymbol{x} + \boldsymbol{e}_j$
7: $\quad$ **if** $g_{k,N}(\boldsymbol{x}^{\text{new}}) > g_{k,N-1}(\tilde{\boldsymbol{x}}^{k,N-1})$ **and** $\boldsymbol{x}^\top \boldsymbol{p} + p_j \leq b_k$ **then**
8: $\quad\quad \mathbb{1}_{\{B_k,D_j,p_j\}} \leftarrow 1$
9: $\quad\quad$ **break**
10: $\quad$ **end if**
11: **end for**
12: **return** $\mathbb{1}_{\{B_k,D_j,p_j\}}$

---

---

**Algorithm 2** Market Dynamic Procedure

---

1: **Inputs:** Buyers $\{B_k\}_{k=1}^M$, Sellers $\{S_j\}_{j=1}^N$
2: **Initialization:** Buyers $\{B_k\}_{k=1}^M$ enter the market
3: **for** $j = 1$ to $N$ **do**
4: $\quad$ Seller $S_j$ enters with potential prices $\mathcal{P}_{j,M}$ for dataset $D_j$
5: $\quad$ **for all** $p_j \in \mathcal{P}_{j,M}$ **do**
6: $\quad\quad$ **for** $k = 1$ to $M$ **do**
7: $\quad\quad\quad$ Buyer $B_k$ solves:

$$\tilde{\boldsymbol{x}}^{k,j-1} = \arg \max_{\boldsymbol{x} \in \mathcal{X}_{k,j}} g_{k,j}(\boldsymbol{x})$$

8: $\quad\quad\quad$ to decide whether to purchase $D_j$ at price $p_j$ $\qquad\qquad$ ▷ See Eqn. 2
9: $\quad\quad$ **end for**
10: $\quad\quad$ Seller computes net profit $r(p_j)$ assuming price $p_j$ $\qquad\qquad$ ▷ See Eqn. 3
11: $\quad$ **end for**
12: $\quad$ Seller selects:

$$p_j^* = \arg \max_{p_j \in \mathcal{P}_{j,M}} r(p_j)$$

13: $\quad$ and sets $p_j^*$ as the fixed price for $D_j$ $\qquad\qquad$ ▷ See Eqn. 4
14: **end for**

---

## F.2 Data Valuation Oracle Experiments

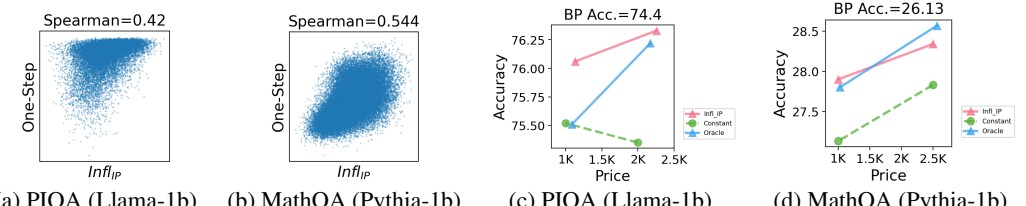

(a) PIQA (Llama-1b) $\qquad$ (b) MathQA (Pythia-1b) $\qquad$ (c) PIQA (Llama-1b) $\qquad$ (d) MathQA (Pythia-1b)

Figure 8: *Left, middle-left columns:* Correlation analysis between oracle and Infl$_{\text{IP}}$ valuation. *Right, middle-right columns:* Performance analysis between between oracle and Infl$_{\text{IP}}$ valuation.

Influence-based methods, such as Infl$_{\text{IP}}$ approximate the influence of a sample $d$ on a $d'$ for a model parameterized by $\theta$ by estimating the effects of training or "upweighting" (alternatively, removing/"downweighting") on $d$ (see Appendix A). In past literature, Infl$_{\text{IP}}$ is validation through an

"Oracle" (One-Step Training) score, which we denote as $\text{Oracle}(d, d') = \mathcal{L}(d'; \theta) - \mathcal{L}(d'; \hat{\theta})$, where $\hat{\theta} = \theta - \eta \nabla \mathcal{L}(d; \theta)$ and $\eta$ is the learning rate [56, 71, 23].

To compare the difference between $\text{Infl}_{\text{IP}}$ valuation versus its oracle valuation, we conduct the same experiments described in Section 4.3. Figure 8 shows that $\text{Infl}_{\text{IP}}$ and Oracle have decent correlation in their agreement in their valuation of the sellers' data, which supports findings in previous works [71]. We note that in the case where correlation is decent, such as in Figure 8b, the final model performance between these methods is close, as seen in Figure 8d. In the case where correlation is lower, such as in Figure 8a, the final model performance between these methods initially have a gap, but become more similar as the amount of data purchased increases, as seen in Figure 8c. This suggests that in practice, even in cases when the agreement between $\text{Infl}_{\text{IP}}$ and Oracle may not be very high, final model performance resulting from these two methods can still be similar.

## F.3    Additional Experimental Results

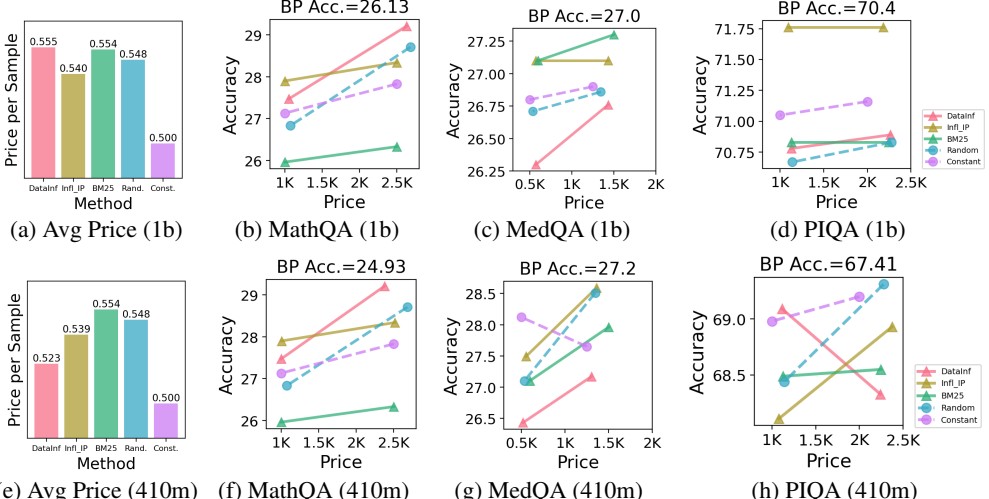

Figure 9: Buyers' model (top row: Pythia-1b, bottom row: Pythia-410m) performance and costs from their purchased data from math, medical, and physical reasoning data markets. Purchasing decisions were using the constant, random, BM25, $\text{Infl}_{\text{IP}}$ data valuation methods (see Section 4.3 for details).

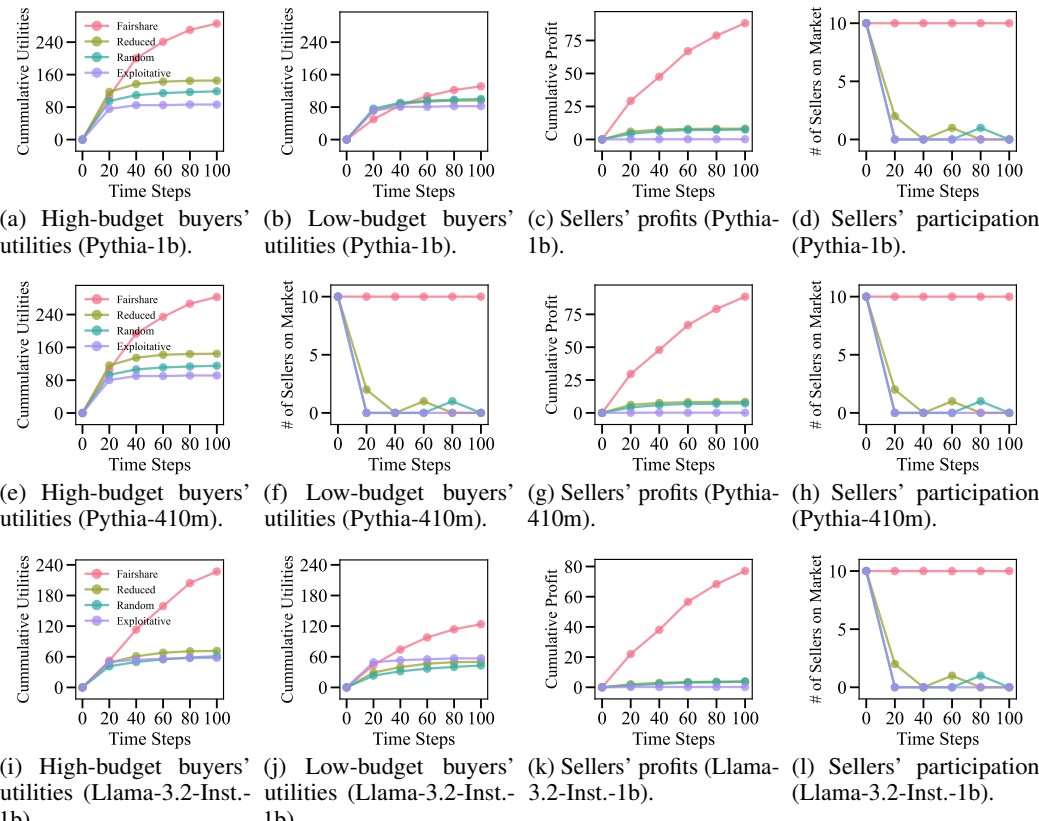

Figure 10: Analysis of (1) buyer's cumulative utilities with high-budget buyer (Figures 10a, 10e and 10i) and low-budget buyer (Figures 10b, 10f and 10j), and (2) sellers' average cumulative profits (Figures 10c, 10g and 10k) and number of sellers in the market (Figures 10d, 10h and 10l) over time ($T = 100$). Model: Pythia-1b, Pythia-410m, and Llama-3.2-Inst.-1b; Task: medqaQA. Experimental groups: (1) fairshare, (2) reduced, (3) random, and (4) current pricing.

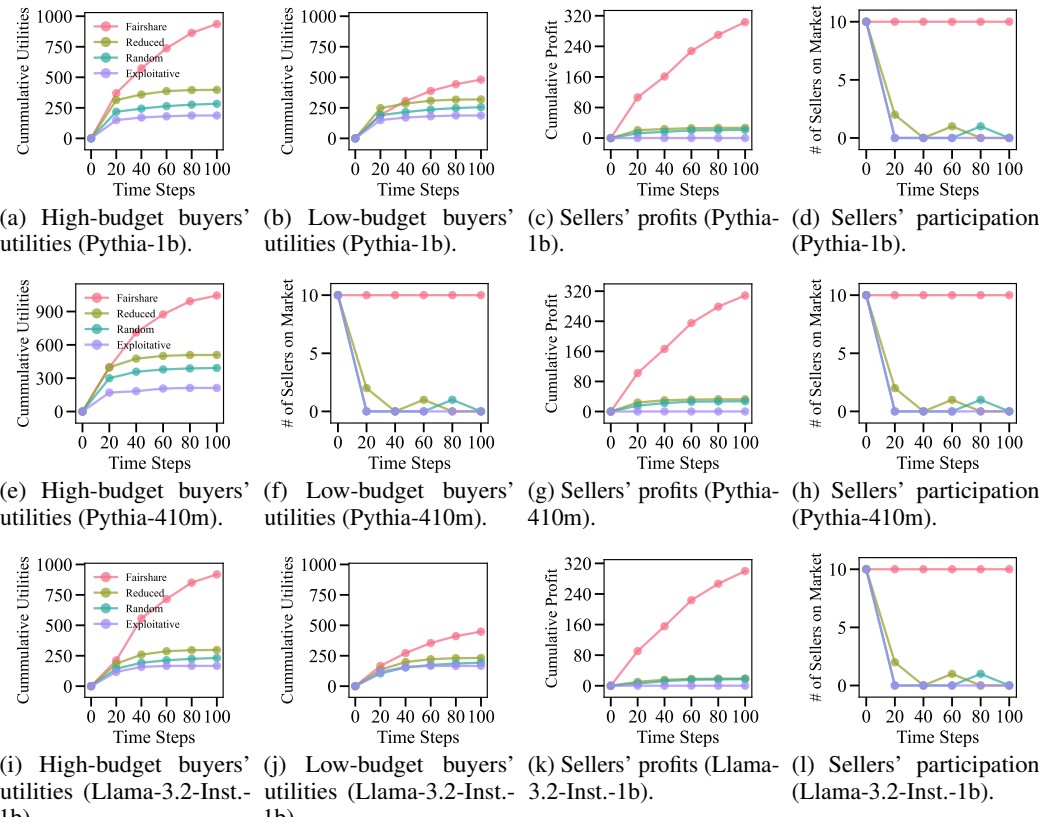

Figure 11: Analysis of (1) buyer's cumulative utilities with high-budget buyer (Figures 11a, 11e and 11i) and low-budget buyer (Figures 11b, 11f and 11j), and (2) sellers' average cumulative profits (Figures 11c, 11g and 11k) and number of sellers in the market (Figures 11d, 11h and 11l) over time ($T = 100$). Model: Pythia-1b, Pythia-410m, and Llama-3.2-Inst.-1b; Task: MathQA. Experimental groups: (1) fairshare, (2) reduced, (3) random, and (4) current pricing.

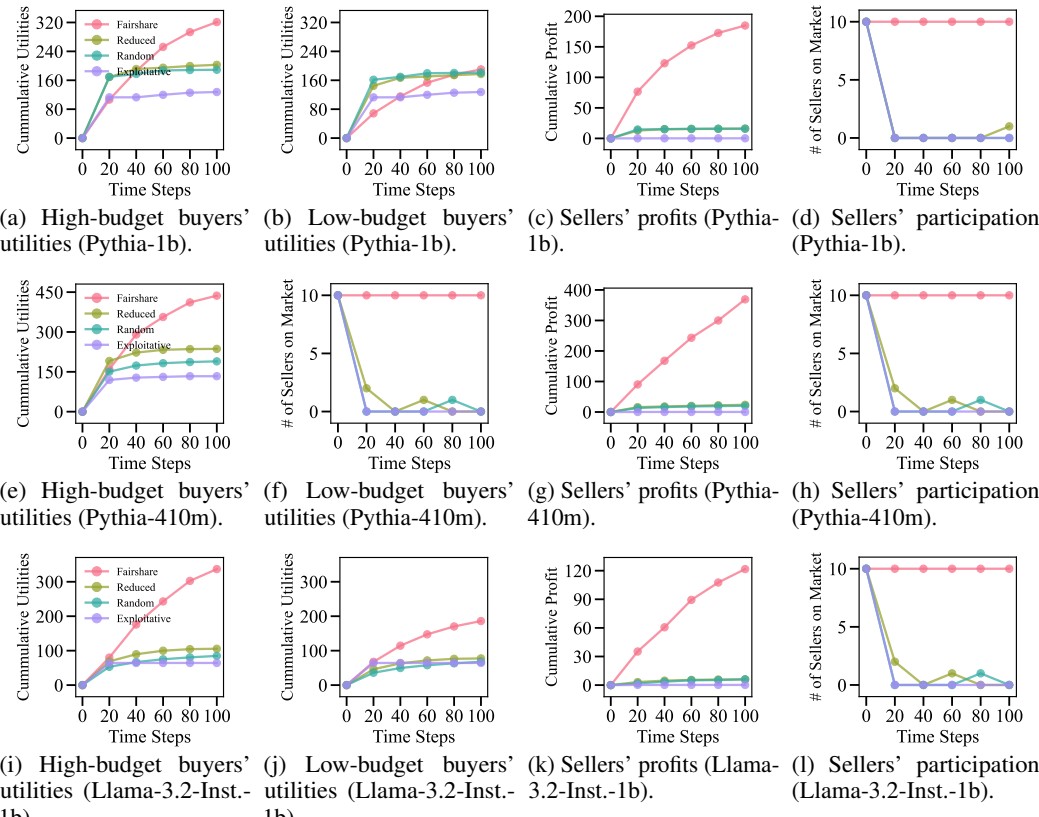

(a) High-budget buyers' utilities (Pythia-1b).

(b) Low-budget buyers' utilities (Pythia-1b).

(c) Sellers' profits (Pythia-1b).

(d) Sellers' participation (Pythia-1b).

(e) High-budget buyers' utilities (Pythia-410m).

(f) Low-budget buyers' utilities (Pythia-410m).

(g) Sellers' profits (Pythia-410m).

(h) Sellers' participation (Pythia-410m).

(i) High-budget buyers' utilities (Llama-3.2-Inst.-1b).

(j) Low-budget buyers' utilities (Llama-3.2-Inst.-1b).

(k) Sellers' profits (Llama-3.2-Inst.-1b).

(l) Sellers' participation (Llama-3.2-Inst.-1b).

Figure 12: Analysis of (1) buyer's cumulative utilities with high-budget buyer (Figures 12a, 12e and 12i) and low-budget buyer (Figures 12b, 12f and 12j), and (2) sellers' average cumulative profits (Figures 12c, 12g and 12k) and number of sellers in the market (Figures 12d, 12h and 12l) over time ($T = 100$). Model: Pythia-1b, Pythia-410m, and Llama-3.2-Inst.-1b; Task: PIQA. Experimental groups: (1) fairshare, (2) reduced, (3) random, and (4) current pricing.

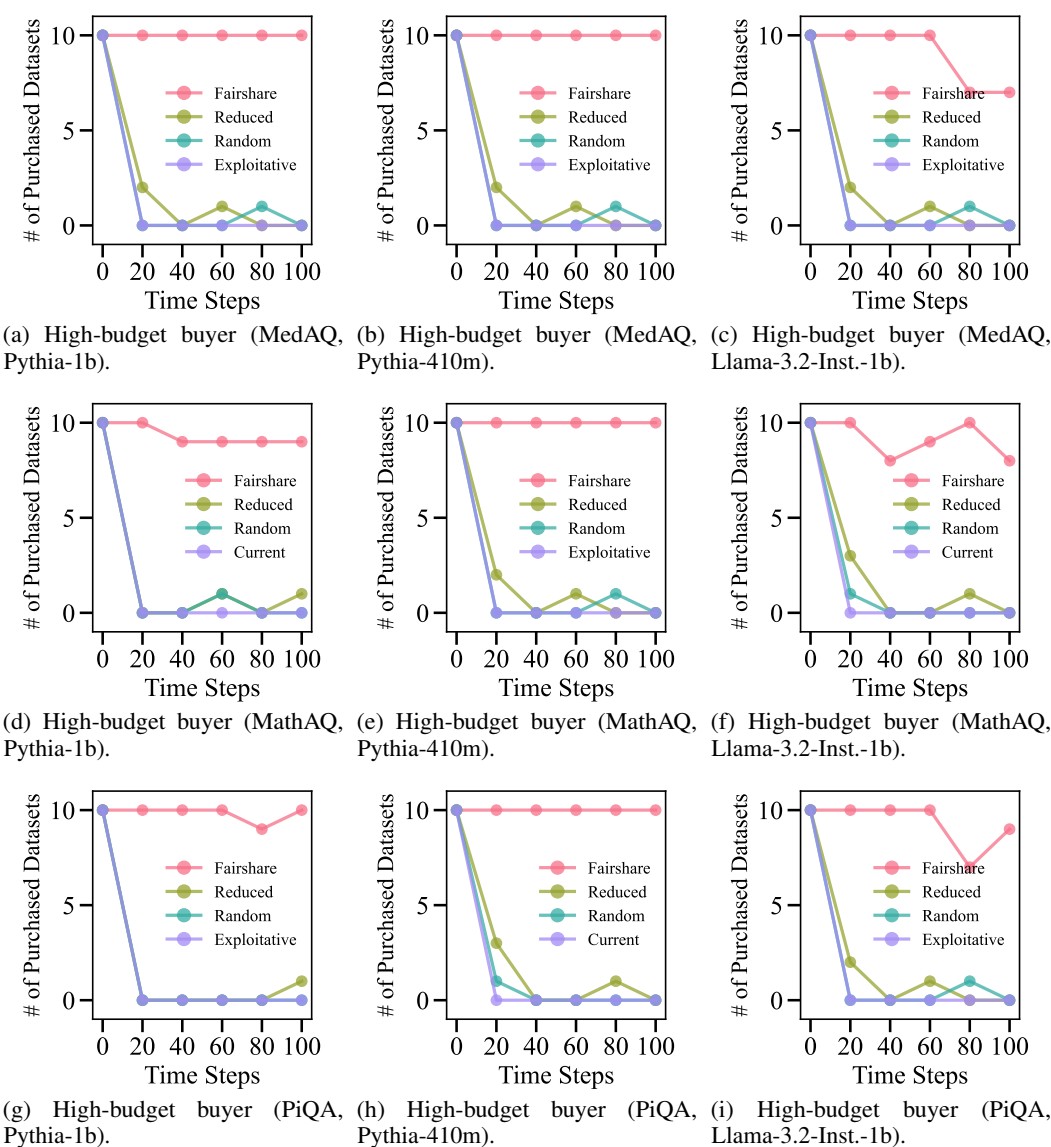

Figure 13: Number of purchased datasets for the buyer with high budget over time periods ($T = 100$). Model: Pythia-1b, Pythia-410m, and Llama-3.2-Inst.-1b; Task: MedQA, MathQA, and PiQA. Experimental groups: (1) fairshare, (2) reduced, (3) random, and (4) current pricing.

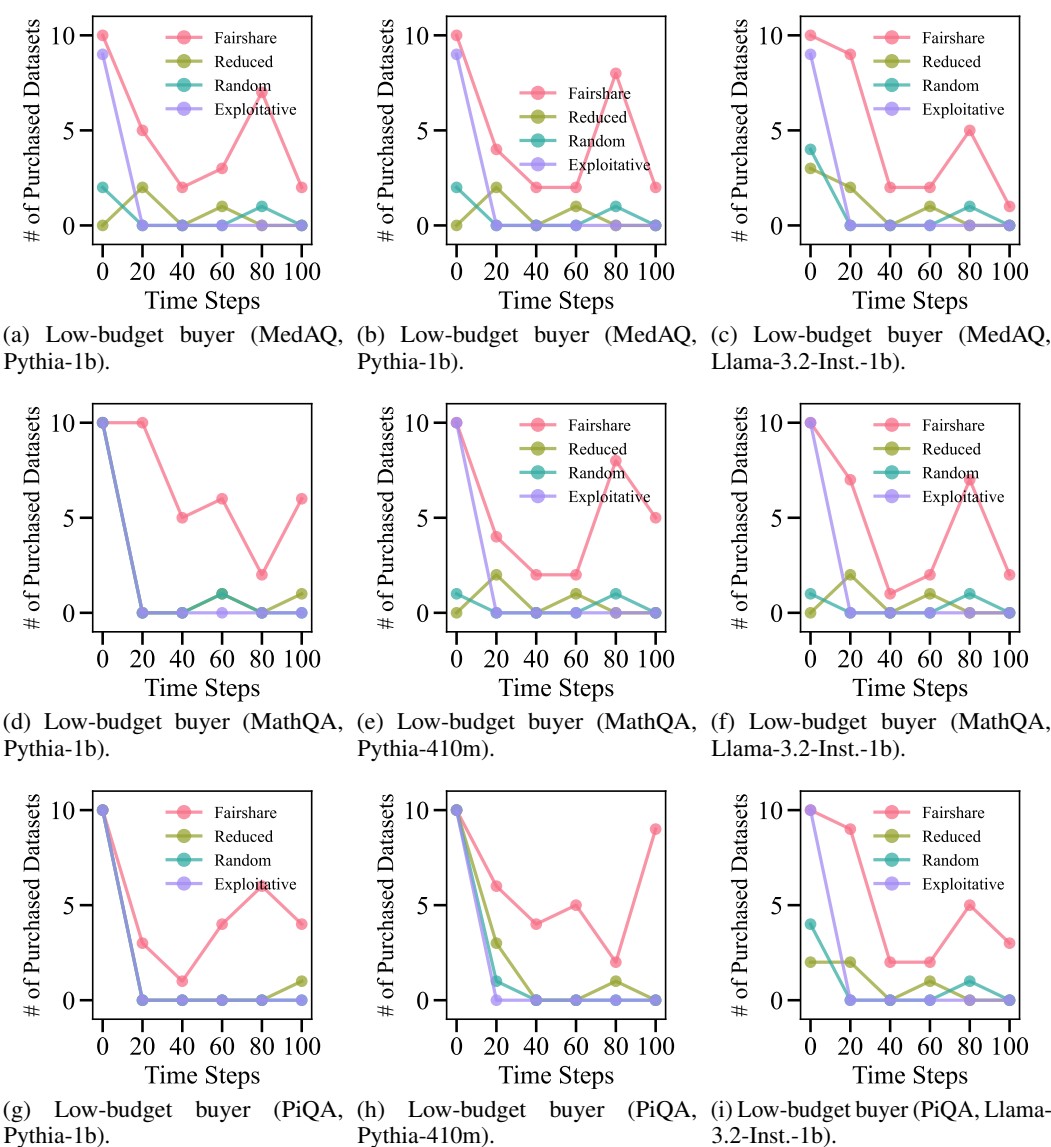

(a) Low-budget buyer (MedAQ, Pythia-1b).

(b) Low-budget buyer (MedAQ, Pythia-1b).

(c) Low-budget buyer (MedAQ, Llama-3.2-Inst.-1b).

(d) Low-budget buyer (MathQA, Pythia-1b).

(e) Low-budget buyer (MathQA, Pythia-410m).

(f) Low-budget buyer (MathQA, Llama-3.2-Inst.-1b).

(g) Low-budget buyer (PiQA, Pythia-1b).

(h) Low-budget buyer (PiQA, Pythia-410m).

(i) Low-budget buyer (PiQA, Llama-3.2-Inst.-1b).

Figure 14: Number of purchased datasets for the buyer with low budget over time periods ($T = 100$). Model: Pythia-1b, Pythia-410m, and Llama-3.2-Inst.-1b; Task: MedQA, MathQA, and PIQA. Experimental groups: (1) fairshare, (2) reduced, (3) random, and (4) current pricing.

## F.4 Datasets

| Dataset | # of Train/Valid/Test | Example |
|---------|----------------------|---------|
| MathQA | 29837/4475/2985 | **Question:** A train running at the speed of 48 km / hr crosses a pole in 9 seconds . what is the length of the train? a ) 140 , b ) 130 , c ) 120 , d ) 170 , e ) 160
**Answer:** C |
| GSM8K | 7473/1319 | **Question:** Natalia sold clips to 48 of her friends in April, and then she sold half as many clips in May. How many clips did Natalia sell altogether in April and May?
**Answer:** 72 |
| MedQA | 10178/1272/1273 | **Question:** A 27-year-old man presents to the emergency room with persistent fever, nausea, and vomiting for the past 3 days. While waiting to be seen, he quickly becomes disoriented and agitated. Upon examination, he has visible signs of difficulty breathing with copious oral secretions and generalized muscle twitching. The patient's temperature is 104°F (40°C), blood pressure is 90/64 mmHg, pulse is 88/min, and respirations are 18/min with an oxygen saturation of 90% on room air. When the nurse tries to place a nasal cannula, the patient becomes fearful and combative. The patient is sedated and placed on mechanical ventilation. Which of the following is a risk factor for the patient's most likely diagnosis? a) Contaminated beef b) Epiglottic cyst c) Mosquito bite d) Spelunking
**Answer:** D |
| PIQA | 16000/2000 | **Question:** How do I ready a guinea pig cage for it's new occupants? a) Provide the guinea pig with a cage full of a few inches of bedding made of ripped paper strips, you will also need to supply it with a water bottle and a food dish. b) Provide the guinea pig with a cage full of a few inches of bedding made of ripped jeans material, you will also need to supply it with a water bottle and a food dish.
**Answer:** A |

Table 1: Dataset splits and demonstrations from the MathQA, GSM8K, MedQA, and PIQA datasets

| Dataset | Prompts |
|---|---|
| MathQA | **Question:** the banker ' s gain of a certain sum due 3 years hence at 10 % per annum is rs . 36 . what is the present worth ? a ) rs . 400 , b ) rs . 300 , c ) rs . 500 , d ) rs . 350 , e ) none of these
**Answer:** A
**Question:** average age of students of an adult school is 40 years . 120 new students whose average age is 32 years joined the school . as a result the average age is decreased by 4 years . find the number of students of the school after joining of the new students . a ) 1200 , b ) 120 , c ) 360 , d ) 240 , e ) none of these
**Answer:** D
**Question:** sophia finished 2 / 3 of a book . she calculated that she finished 90 more pages than she has yet to read . how long is her book ? a ) 229 , b ) 270 , c ) 877 , d ) 266 , e ) 281
**Answer:** B
**Question:** 120 is what percent of 50 ? na ) 5 % , b ) 240 % , c ) 50% , d ) 2 % , e ) 500
**Answer: B**
**Question:** there are 10 girls and 20 boys in a classroom . what is the ratio of girls to boys ? a ) 1 / 2 , b ) 1 / 3 , c ) 1 / 5 , d ) 10 / 30 , e ) 2 / 5
**Answer:** A |
| MedQA | **Question:** A mother brings her 3-week-old infant to the pediatrician's office because she is concerned about his feeding habits. He was born without complications and has not had any medical problems up until this time. However, for the past 4 days, he has been fussy, is regurgitating all of his feeds, and his vomit is yellow in color. On physical exam, the child's abdomen is minimally distended but no other abnormalities are appreciated. Which of the following embryologic errors could account for this presentation? a) Abnormal migration of ventral pancreatic bud b) Complete failure of proximal duodenum to recanalize c) Abnormal hypertrophy of the pylorus d) Failure of lateral body folds to move ventrally and fuse in the midline
**Answer:** A
**Question:** A 53-year-old man comes to the emergency department because of severe right-sided flank pain for 3 hours. The pain is colicky, radiates towards his right groin, and he describes it as 8/10 in intensity. He has vomited once. He has no history of similar episodes in the past. Last year, he was treated with naproxen for swelling and pain of his right toe. He has a history of hypertension. He drinks one to two beers on the weekends. Current medications include amlodipine. He appears uncomfortable. His temperature is 37.1˚C (99.3˚F), pulse is 101/min, and blood pressure is 130/90 mm Hg. Examination shows a soft, nontender abdomen and right costovertebral angle tenderness. An upright x-ray of the abdomen shows no abnormalities. A CT scan of the abdomen and pelvis shows a 7-mm stone in the proximal ureter and grade I hydronephrosis on the right. Which of the following is most likely to be seen on urinalysis? a) Urinary pH: 7.3 b) Urinary pH: 4.7 c) Positive nitrites test d) Largely positive urinary protein
**Answer:** B
**Question:** A 48-year-old woman comes to the emergency department because of a photosensitive blistering rash on her hands, forearms, and face for 3 weeks. The lesions are not itchy. She has also noticed that her urine has been dark brown in color recently. Twenty years ago, she was successfully treated for Coats disease of the retina via retinal sclerotherapy. She is currently on hormonal replacement therapy for perimenopausal symptoms. Her aunt and sister have a history of a similar skin lesions. Examination shows multiple fluid-filled blisters and oozing erosions on the forearms, dorsal side of both hands, and forehead. There is hyperpigmented scarring and patches of bald skin along the sides of the blisters. Laboratory studies show a normal serum ferritin concentration. Which of the following is the most appropriate next step in management to induce remission in this patient? a) Pursue liver transplantation b) Begin oral thalidomide therapy c) Begin phlebotomy therapy d) Begin oral hydroxychloroquine therapy
**Answer:** C
**Question:** A 23-year-old pregnant woman at 22 weeks gestation presents with burning upon urination. She states it started 1 day ago and has been worsening despite drinking more water and taking cranberry extract. She otherwise feels well and is followed by a doctor for her pregnancy. Her temperature is 97.7˚F (36.5˚C), blood pressure is 122/77 mmHg, pulse is 80/min, respirations are 19/min, and oxygen saturation is 98% on room air. Physical exam is notable for an absence of costovertebral angle tenderness and a gravid uterus. Which of the following is the best treatment for this patient? a) Ampicillin b) Ceftriaxone c) Doxycycline d) Nitrofurantoin
**Answer:** D |

Table 2: Demonstrations included for 5-shot evaluation on the MathQA dataset and for 4-shot evaluation on the MedQA dataset. Demostrations were randomly selected from their respective dataset's training sets.

# G   Limitations and Impact

This paper addresses the critical issue of fairshare pricing in the data market for large language models (LLMs) by proposing a framework and methodologies for fair compensation of datasets from LLM developers to data annotators. Our work directly tackles the ethical and societal challenges in the current data market, where many data annotators are underpaid and receive compensation significantly disconnected from the true economic value their contributions bring to LLMs.

## G.1   Limitations

Since our work proposes a novel fairshare framework, there are several lines of future research that can investigate future adjustments to this framework, which lie beyond the scope of our paper. For instance, a large-scale simulation of this market with a wider range of datasets and models is one possibility. In addition, running the simulation with human buyers/sellers is another avenue. Finally, there are several other diverse market dynamics (e.g., incomplete information between buyers/sellers) that can be explored with our proposed framework.

## G.2   Impact Statement

From ethical and societal perspectives, our framework prioritizes the welfare of both data annotators and LLM developers. Our methodology ensures that data annotators are fairly compensated for their labor, promoting equity and fairness in the data ecosystem. This contributes to mitigating the exploitation of vulnerable annotators in the data market and aligns the incentives of stakeholders toward a more ethical and sustainable practice. In addition, our framework also benefits LLM developers, by demonstrating that our framework maximizes their utilities and welfare in the long term. Fair compensation encourages ongoing participation of data annotators in the market, ensuring a steady supply of diverse, high-quality datasets essential for LLM development. By addressing existing inequities, our work lays the foundation for a more sustainable, equitable, and mutually beneficial ecosystem for all stakeholders in the LLM data market.

