# OpenReview forum: "Fairshare Data Pricing via Data Valuation for Large Language Models"
_NeurIPS.cc/2025/Conference — NeurIPS 2025 poster_

### Official Review · Reviewer_nL1m · 2025-07-02

**Clarity:** 3
**Significance:** 3
**Originality:** 3
**Rating:** 5
**Confidence:** 3

**Summary:**

This paper proposes a theoretical framework to model the LLM data market, where buyers and sellers make decisions with transparent information. Theoretical and simulation results show that exploitative pricing leads to a lose-lose situation where data sellers exit the market, leaving buyers with data shortage. Furthermore, the authors propose a fairshare pricing mechanism that theoretically and empirically lead to a win-win situation, where data sellers obtain optimal profits and the buyers obtain long-run utility. Finally, across three LLMs, four tasks, and two budgets, the proposed fairshare pricing achieves the win-win situation as the theory suggests. These empirical results are consistent across several data valuation methods that are scalable to LLMs.

**Questions:**

- Intuitively, Assumptions 1 and 1.1 suggest that the sellers are quite likely to exit the market once they are exploited. Can the authors comment on how realistic this is in the real world? What if the lower bound in Assumption 1.1 is replaced with an upper bound, and $P_T$ in Assumption 1 is not multiplicative, suggesting that sellers are more tolerant to exploitation?
- In Figure 2(c), at Buyer 1's MWP, why is the profit (~2.0) larger than the price (<2.0)?
- This is a minor comment. Should $R_{\ge 0}$ in line 139 be replaced by $R$ for the range of the data valuation function?

**Ethical Concerns:**

["NO or VERY MINOR ethics concerns only"]

**Final Justification:**

I keep the initially recommended score of 5. My primary concerns were about (1) a lack of sufficient assumptions for Lemma 2 and (2) a lack of discussion about what happens when the assumptions made in the paper break. The authors have addressed my concerns with their rebuttal. While I agree with Reviewer 9cVB that the assumed seller behavior can be unrealistic, I view this paper as a reasonable first step toward formalizing data pricing in the LLM data market and initiating formal discussion.

**Limitations:**

The limitations seem adequately addressed.

**Paper Formatting Concerns:**

None.

**Quality:**

3

**Strengths And Weaknesses:**

**Strengths**
- This work formalizes the LLM data market with a theoretical framework.
- This work proposes a fairshare pricing mechanism based on the theoretical framework. This can be impactful for policymaking and the fairness domain.
- The lose-lose outcome from exploitative pricing and the win-win outcome of the proposed fairshare pricing are well supported with theoretical and simulation results, as well as empirical results with actual LLM fine-tuning.

**Weaknesses**
- The seller-side pricing under fairshare pricing (Section 3.3) does not seem to take the cost $c_j$ into consideration, which is inconsistent with the overall market framework. In particular, $p^*_j$ may be outside $\cup_k MWP_k$ if $c_j > MWP_k$ for all $k = 1, ..., M$. I think a minor assumption on $c_j$ is needed for Lemma 2 to hold. Intuitively, $c_j$ should not be larger than a buyer's budget and marginal utility.
- Overall, Assumptions 1, 1.1, and 2 are reasonable. However, the authors should comment on how their results would change if these assumptions are broken.

---

> ### Author Rebuttal · Authors · 2025-07-30
>
> We thank Reviewer nL1m for their highly constructive feedback and insightful technical questions. The suggestions and feedback by Reviewer nL1m are extremely valuable for this paper. We appreciate the opportunity to clarify these points and address them below.
>
> **1. On the Role of Cost cj​ in Fairshare Pricing (Lemma 2):**
>
> We thank the reviewer for this sharp and accurate observation. The reviewer is correct that our characterization of the optimal price in Lemma 2 implicitly assumes that a profitable transaction is possible for the seller (i.e., $p_{j}^{*} \geq c_{j}$). We will make this assumption explicit in Section 3.3 to improve the rigor of our claims. We appreciate the reviewer helping us strengthen our presentation.
>
> We would also like to briefly clarify the justification for this assumption. It is grounded in the current dynamics of the LLM data market. Companies are making significant investments to acquire high-quality training data, as its value is a critical driver of model performance. On the other hand, the labor cost to produce this data is often exploitatively low [5-8]. This makes it a reasonable assumption for our model that the optimal price a buyer is willing to pay for valuable data ($p_{j}^{*}$) generally exceeds its annotation and production cost ($c_{j}$).
>
> **2. On the Justification of Assumptions 1, 1.1:**
>
> We thank the reviewer for this excellent and nuanced question, which gets to the heart of our behavioral model. Our assumption that sellers are sensitive to exploitation is not just an intuition but is strongly grounded in both classical economic theory and direct evidence from the data annotation market.
>
> First, our model aligns with foundational studies in organizational justice and fairness theory, which we cite in our Related Work section. This body of literature shows that worker retention is driven by perceived fairness [1-4], not just absolute payment. When compensation is seen as disconnected from the value created – a core feature of exploitative pricing – it erodes trust and motivation, reducing the willingness of skilled participants to remain in a market. This principle has been documented specifically in the "ghost work" economy, where precarious conditions and low pay lead to high worker churn and disengagement. This has been highlighted in our Related Work (line 104-105).
>
> Second, this long-standing theoretical principle is being validated in real-time in the current LLM data market. Recent investigations by outlets like WIRED [5] and The Washington Post [6] describe the industry as experiencing high worker churn due to persistently low and inconsistent pay. More directly, widespread reports from early 2024 revealed that when platforms like Scale AI's Remotasks abruptly reduced pay rates, it led to an immediate exodus of skilled annotators [7]. Further, a significant 2025 investment in Scale AI by Meta has raised concerns about gig workers being left behind, underscoring ongoing exploitation concerns [8]. These reports provide direct, contemporary evidence for our model's core assumption: exploitative pay strategies lead to high dropout rates among the very workers needed for high-quality data. We will incorporate these contemporary reports into our revised manuscript to further strengthen the real-world grounding for our model's behavioral assumptions.
>
> **3. On the Violation of Assumptions 1, 1.1, and 2:**
>
> We thank the reviewer for this excellent suggestion. It is important to consider the boundaries of our model. If they were violated, the dynamics would change as follows:
>
> **If Assumptions 1 and 1.1 are broken:** We thank the reviewer for pointing this out and fully agree on the reviewer’s intuition. This implies that sellers are not sensitive to exploitative pricing and are highly tolerant of being underpaid. In this case, the primary long-term penalty for the buyer (i.e., losing access to high-quality data suppliers) is removed.
>
> **If Assumption 2 is broken:** This means the buyer is myopic and short-sighted, caring only about maximizing their immediate surplus ($u_t - p_t$) rather than their cumulative long-term utility.
>
> In a scenario where both sets of assumptions are violated, the incentive structure that upholds our fairshare equilibrium might collapses. With tolerant sellers and a short-sighted buyer, the market would be dominated by exploitative pricing, as the buyer would have no rational incentive to pay more for long-term market sustainability. We will add a discussion to Section 3 analyzing how our results would change if these core assumptions were relaxed, making the scope of our framework clearer.
>
> **5. Clarification of Figure 2**
>
> We thank the reviewer for this careful observation of Figure 2(c). The price point labeled "Buyer 1's MWP" is the highest price at which both buyers are willing to purchase the dataset. Therefore, the seller's total profit at this point is approximately (2 * price) - cost, which is why the profit value (~2.0) can be larger than the price itself (<2.0). If the seller were to increase the price beyond this point, Buyer 1 would drop out, leading to a lower total profit. We will revise the caption of Figure 2 to explicitly state that the simulation involves two buyers and that the plot shows the seller's total profit to ensure this is clear.
>
> **4. The Range of Domain:**
>
> Thank you for your careful reading. The reviewer is correct. We will update the notation in line 139 from $R\geq0$​ to $R$ for full generality. We thank the reviewer again for their supportive assessment and the detailed feedback, which will improve the rigor and clarity of our paper.
>
> ***
>
> [1] Robert G Folger, Robert Folger, and Russell Cropanzano. *Organizational justice and human resource management, volume 7.* 1998.
>
> [2] Robert Folger and Russell Cropanzano. *Fairness theory: Justice as accountability.* Advances in organizational justice. 2001.
>
> [3] Jason A Colquitt, Jerald Greenberg, and Cindy P Zapata-Phelan. *What is organizational justice? a historical overview. In Handbook of organizational justice.* Psychology Press, 2013.
>
> [4] Mladen Adamovic. *Organizational justice research: A review, synthesis, and research agenda.* European Management Review. 2023.
>
> [5] Niamh Rowe. *Millions of Workers Are Training AI Models for Pennies.* WIRED. 2023
>
> [6] Rebecca Tan and Regine Cabato. *Behind the AI boom, an army of overseas workers in ‘digital sweatshops’.* The Washington Post.  2023.
>
> [7] Russell Brandom. *Scale AI’s Remotasks platform is dropping whole countries without explanation.* Rest of World. 2024
>
> [8] Billy Perrigo. *Meta’s $15 Billion Scale AI Deal Could Leave Gig Workers Behind.* TIME. 2025.

---

> > ### Comment · Reviewer_nL1m · 2025-08-06
> >
> > I appreciate the authors for the rebuttal. All my concerns and questions have been addressed. I recommend the authors to include in their paper how the current LLM data market aligns with the assumptions discussed in the rebuttal.

---

> > > ### Author Response · Authors · 2025-08-06
> > >
> > > We sincerely thank the reviewer for the constructive review and strong support. That is an excellent final suggestion. We absolutely agree that adding the discussion on how our assumptions align with the current LLM data market will greatly improve the paper, and we will be sure to incorporate it into the next version. Thank you again for your valuable feedback.

---

### Official Review · Reviewer_AWaS · 2025-07-03

**Clarity:** 3
**Significance:** 3
**Originality:** 4
**Rating:** 5
**Confidence:** 3

**Summary:**

Authors propose a theoretical fairshare data pricing mechanism that maximizes the seller's profits while aligning with the dataset's utility and the buyer's budget.
The mechanism relies on data valuation methods to determine the value of data, and the authors use data for training LLMs as a motivating example, theoretically and empirically.
The authors also empirically test their framework in a semi-synthetic setting involving two buyers, three open-source LLMs, and 10 sellers.

**Questions:**

- Mismatch between motivation and proposed framework.
  - Although the proposed model is technically sound, it doesn't match the motivation of the work. The authors highlight the ethical concerns in LLM data acquisition, for example, the Sama company, Kenyan workers, and the OpenAI case ([11,12]). Accordingly to the theoretical framework, the sellers are actually companies like Sama, and not the Kenyan annotators. Additionally, the authors mention that the sellers would stop getting involved in the data market if not fairly treated, but as it was in this case, they simply move to new sellers because there is always someone willing to. So entry and exit of data buyers and sellers in the market should ideally be covered.
  - There might be a decrease in data quality at time t, but a rise in quality at time t+x due to unfair data market or other factors outside of this. The model doesn't account for the changes in data quality at various time steps or for the entry and exit of different sellers and buyers in the data market, due to exploitative pricing and other factors.

- Sections 3.1 and 3.2 and Appendix A, B and C
  - I found Figure 1 confusing and unintuitive. For example, the data valuation methods require a model performance to compute the value of data, which is not clear from the figure. The relationship between the seller and buyer is oversimplified. For example, from the follow-up text, it seems to me there might be a back and forth between the two, sellers need potentially broadcasted buyers decision-making, and buyers need the data to compute value and set the price - to make decisions.
  - The model of the seller's decision-making (Eqs. 3 and 4) has strong assumptions. For example, a lay seller, e.g., an annotator, might not have access to the seller utility or other data sources. As such, computing a net profit function is not trivial, and sellers might over- or underprice their data. Also how do we know the fixed price (line 158) is fair? Can buyers only get a fraction of seller data (Section 3.1 and Appendix B)?
  - In lines 141 to 144, the authors show that utility (not the usual language of utility in dval literature) is a function of data values. It's unclear to me how well or accurately this formulation of utility captures gains such as commercial returns that buyers care about (line 196).   - The utility formulation has other limitations. In line 674, how high is high?  What's the threshold for determining that? All the utility formulations (C.1 to C.3) rely on marginal contribution-based data valuation methods defined in lines 141. I imagine it could be possible to combine various forms of valuating data to determine the utility.
  - The setup for the results in Figure 3 is oversimplified. For example, why is utility u constant?

- Experimental setups and analyses
  - Given that authors consider varied tasks, why do they only use the affine mappings and not test the other formulations of utility? (c.2, c.3, c.4)?
  - There is little justification or reasoning given for the choice of various parameters, e.g., how did the authors choose \delta and T?

- Limitations or discussions section
  - Given that the authors make so many assumptions and the framework could be understood from various viewpoints, I think the paper should have a limitations section or discussions to include some of these perspectives and assumptions.

**Ethical Concerns:**

["NO or VERY MINOR ethics concerns only"]

**Final Justification:**

The authors have addressed my concerns and those of other reviewers. Despite some weaknesses, the paper presents a novel and scalable idea, supported by theoretical analysis and empirical results. I have therefore decided to slightly increase my score, with the expectation that the authors will incorporate the revisions they have committed to in the rebuttals, such as improving the introduction, refining Section 3, and adding a more comprehensive limitations section.

**Limitations:**

Given that the authors make so many assumptions and the framework could be understood from various viewpoints, I think the paper should have a limitations section or discussions to include some of these perspectives and assumptions.

**Paper Formatting Concerns:**

No concerns

**Quality:**

2

**Strengths And Weaknesses:**

**Strengths**
- The authors provide a nice background and motivation for their proposed framework of data pricing.
- The theoretical model is well-motivated, technically sound, clearly written, and easy to parse. I especially like the authors' idea of incorporating both sellers' and buyers' utility and the long-term effects of (un)fair data pricing.
- The motivating empirical results in Figures 2 and 3 are intuitive and complementary to their setup.
- In the empirical experiments, the authors follow up with simulations specific to the LLM running example.

**Weaknesses**

- Mismatch between motivation and proposed framework.
  - Although the proposed model is technically sound, it doesn't match the motivation of the work. The authors highlight the ethical concerns in LLM data acquisition, for example, the Sama company, Kenyan workers, and the OpenAI case ([11,12]). Accordingly to the theoretical framework, the sellers are actually companies like Sama, and not the Kenyan annotators. Additionally, the authors mention that the sellers would stop getting involved in the data market if not fairly treated, but as it was in this case, they simply move to new sellers because there is always someone willing to. So entry and exit of data buyers and sellers in the market should ideally be covered.
  - There might be a decrease in data quality at time t, but a rise in quality at time t+x due to unfair data market or other factors outside of this. The model doesn't account for the changes in data quality at various time steps or for the entry and exit of different sellers and buyers in the data market, due to exploitative pricing and other factors.

- Sections 3.1 and 3.2 and Appendix A, B and C
  - I found Figure 1 confusing and unintuitive. For example, the data valuation methods require a model performance to compute the value of data, which is not clear from the figure. The relationship between the seller and buyer is oversimplified. For example, from the follow-up text, it seems to me there might be a back and forth between the two, sellers need potentially broadcasted buyers decision-making, and buyers need the data to compute value and set the price - to make decisions.
  - The model of the seller's decision-making (Eqs. 3 and 4) has strong assumptions. For example, a lay seller, e.g., an annotator, might not have access to the seller utility or other data sources. As such, computing a net profit function is not trivial, and sellers might over- or underprice their data. Also how do we know the fixed price (line 158) is fair? Can buyers only get a fraction of seller data (Section 3.1 and Appendix B)?
  - In lines 141 to 144, the authors show that utility (not the usual language of utility in dval literature) is a function of data values. It's unclear to me how well or accurately this formulation of utility captures gains such as commercial returns that buyers care about (line 196).   - The utility formulation has other limitations. In line 674, how high is high?  What's the threshold for determining that? All the utility formulations (C.1 to C.3) rely on marginal contribution-based data valuation methods defined in lines 141. I imagine it could be possible to combine various forms of valuating data to determine the utility.
  - The setup for the results in Figure 3 is oversimplified. For example, why is utility u constant?

- Experimental setups and analyses
  - Given that authors consider varied tasks, why do they only use the affine mappings and not test the other formulations of utility? (c.2, c.3, c.4)?
  - There is little justification or reasoning given for the choice of various parameters, e.g., how did the authors choose \delta and T?

- Limitations or discussions section
  - Given that the authors make so many assumptions and the framework could be understood from various viewpoints, I think the paper should have a limitations section or discussions to include some of these perspectives and assumptions.

---

> ### Author Rebuttal · Authors · 2025-07-31
>
> We thank Reviewer AWaS for their detailed and thoughtful review. We address these points below.
>
> **1. Mismatch Between Motivation and Framework:**
>
> We thank the reviewer for this insightful observation. It is absolutely correct that the real-world data supply chain often involves a three-party structure: Buyer, Intermediary, and Supplier. Our choice to model a two-party (Buyer-Seller) dynamic is a deliberate one, intended as a crucial first step. Importantly, this model directly represents many real-world markets, such as major crowdsourcing platforms (e.g., Amazon MTurk and Prolific), where buyers interact directly with sellers (annotators).
>
> Our “seller” is a tractable abstraction of the data supply side. The exploitative dynamic we model is often amplified in a three-party reality with an intermediary, a phenomenon explained by the economic literature on intermediation [1,3]. Specifically, the concept of "double marginalization" [2,3] shows how an intermediary's need to secure its own profit margin creates intense downward pressure on annotator wages to satisfy a price-sensitive buyer.
>
> The fundamental conclusion of our work (i.e., under-compensation harms long-term data quality) remains the central tension in both two-party and three-party settings. Our current framework is valuable because it is a tractable model that also directly represents a large fraction of the data market, such as major crowdsourcing platforms. Modeling the additional strategic layer of an intermediary is a crucial and complex issue that demands future research, and we hope our work provides a clear foundation for these investigations.
>
> Our two-party model serves as a tractable and powerful abstraction that isolates this core economic conflict. Formally modeling a three-party game, while a fascinating direction for future work, would introduce significant additional complexities (e.g., contract design, monitoring costs, three-way information asymmetry) that would obscure the foundational insight we aim to establish. We believe it is standard and valuable to first understand the core dyad before modeling the full supply chain.
>
> We will revise the introduction and Section 3 to explicitly acknowledge the three-party structure of the real-world market. We will state that our two-party model is a deliberate abstraction and will briefly introduce the concept of "double marginalization" to argue that our model's core conclusions are likely amplified, not invalidated, in this more complex setting. We will also add a formal three-party game as a key direction for future work in our conclusion.
>
> **2. Exit of Buyers/Sellers:**
>
> These are excellent points. Even when new sellers can enter the market, the core principle of our work remains the same: underpayment is a self-defeating strategy. This is because, in a transparent market, an exploitative buyer would develop a poor reputation, deterring new, high-quality sellers and leading to a "lemons market" of suppliers. Furthermore, constantly replacing exiting sellers is not a sustainable strategy but a high-cost, inefficient cycle of churn [4]. While formally modeling these entry/exit and reputation dynamics is a key direction for future work, we will add this discussion to our limitations section to scope our contribution and highlight this as an important avenue for research.
>
> **3. Change of Data Quality:**
>
> We thank the reviewer for these insightful questions. Our framework is intentionally designed to be agnostic to the specific form of the utility function. Thus, it can readily accommodate these complexities, which represent key directions for future work. Fluctuating data quality can be directly modeled by treating the buyer's utility as a stochastic variable; our core conclusions still hold by simply using the expected utility in the decision-making process. We chose a deterministic setup in the paper for clarity and to crisply isolate the core strategic trade-offs, but the framework's ready extension to stochastic cases demonstrates its robustness.
>
> **4. On Unclear Figures and Simplified Setups:**
>
> We thank the reviewer for this helpful feedback.  **Figure 1:** We will revise it and its caption to better illustrate the role of data valuation and the sequential buyer-seller interaction. **Figure 3:** This is a simplified illustration to clearly visualize our theoretical result in Lemma 3. We used a fixed utility to isolate the long-term strategic trade-off from stochastic noise.  In Section 4, our main experiments adopt stochastic utilities. We will revise Figure 3 to make its illustrative purpose explicit.
>
> **5. Complete Information Assumption:**
>
> We thank the reviewer for raising this important point. Our choice to model a transparent market, where a platform provides public data valuation scores. This operationalizes information transparency and makes the problem tractable. Our choice to model this setting first was deliberate for three reasons:
>
> 1. It serves as an essential first step to isolate the core economic interaction between buyers and sellers before introducing the complexities of asymmetric information.
>
> 2. It follows the established precedent of seminal works in economics such as those by Akerlof [4] and Spence [5], which first established key insights in simpler settings. And follow-ups build the pipeline throughout more complex settings.
>
> 3. Modeling incomplete information is a vast future research direction, and our work provides the critical first step.
>
> This assumption also aligns with behavioral fairness models that highlight transparency’s role in sustaining cooperation [4,5,7]. We will further clarify in Section 3 that the seller's problem is made tractable by public valuation signals and emphasize how this benchmark lays the groundwork for future study of incomplete information settings.
>
> **6. Utility Function Mapping:**
>
> We thank the reviewer for excellent questions about utility. Our framework was designed with generality and modularity in mind, allowing it to be adapted to diverse real-world scenarios.
>
> **Utility and Commercial Returns:** We use “utility” in its classic economic sense as a flexible measure of value [6,7]. Our framework is modular, separating the valuation score $v_k$ (technical impact) from a buyer-defined utility mapping $u_k(v_k)$. This allows practitioners to translate technical impact (i.e., model performance) into firm-specific value, such as commercial returns. This allows our theoretical framework to remain general, while letting practitioners plug in their own specific goals.
>
> **Thresholds:** The specific mappings in Appendix C are illustrative of this flexibility. Defining empirically-grounded thresholds is a firm-specific implementation detail that our framework is designed to accommodate, leaving this as a direction for future applied work with empirical data.
>
> **Combining Valuation Methods:** Absolutely agree. The framework is agnostic to how the valuation score $v_k$ is derived. A buyer can use a composite score (e.g., an average or minimum of various methods) for more robust signals, and our conclusions still hold.
>
> We will revise Section 3 to clarify that the utility mapping is a modular, buyer-defined component, highlighting the generality of our framework.
>
> **8. On Experimental Setups and Analyses:**
>
> We thank the reviewer for this excellent question.
>
> **Other utility mappings:** In our main experiments, we used the affine mapping (Appendix C.1) primarily for its clarity and direct interpretability. Our primary reason we did not include experiments on all mappings from Appendix C is that our framework is designed to be method-agnostic. The conclusions are expected to hold for other formulations.
>
> To empirically validate this, we run new experiments using the Discrete Outcome mapping (Appendix C.2). The results are highly consistent with those in the main paper: Fairshare pricing continues to maximize long-term utility for buyers and profits for sellers. This confirms that our central conclusions are robust to the specific formulation of utility.
>
> **Cumulative Buyer's Utility (Discrete Mapping)**
> | Time Steps | Fairshare | Discount | Random | Exploitative |
> |---|---|---|---|---|
> | 20 | 51.11 | 115.66 | 90.98 | 72.87 |
> | 60 | 146.70 | 155.62 | 124.75 | 87.08 |
> | 100 | 236.49 | 162.67 | 134.55 | 99.53 |
>
> **Choice of $\delta$ and $T$:** Our selections are grounded in standard practice and validated within our paper.
>
> $\delta$: Following standard approaches in dynamic programming and economics [8,9], we set $\delta=0.98$ to ensure future outcomes are properly considered and not overly discounted. This aligns with our Assumption 2, reflecting forward-looking buyers.
>
> $T$: 100 steps adequately capture the long-term market dynamics central to our analysis, allowing stable patterns and equilibrium outcomes to emerge.
>
> We also thoroughly analyze the impact of these parameters. In Lemma 7 (line 791), we theoretically prove the trade-off, showing that a less patient buyer (a smaller $\delta$) requires a longer time to converge to the optimal fairshare strategy. To complement this, we empirically demonstrate this exact relationship in our robustness check in Appendix E.2.
>
> **10. Limitations and discussions:**
>
> We thank the reviewer for this important suggestion. While our current limitations section in Conclusion and Appendix G already encourages future work on incomplete information settings, we will add a more thorough discussion in the next version.
> ***
> [1]Coase.The Nature of the Firm.
>
> [2]Spengler.Vertical Integration and Antitrust Policy.
>
> [3]Tirole.The Theory of Industrial Organization.
>
> [4]Akerlof.The market for “lemons”: Quality uncertainty and the market mechanism.
>
> [5]Spence.Job market signaling.
>
> [6]Mas-Colell et al.Microeconomic theory.
>
> [7]Neumann et al.Theory of games and economic behavior.
>
> [8]Sutton et al.Reinforcement Learning: An Introduction.
>
> [9]Stokey et al.Recursive Methods in Economic Dynamics.

---

> > ### Comment · Reviewer_AWaS · 2025-08-03
> >
> > Thank you to the authors for addressing my questions. I have also reviewed comments by other reviewers as well as the authors’ rebuttals.
> >
> > Despite some weaknesses, the paper presents a novel and scalable idea, supported by theoretical analysis and empirical results.
> >
> > In light of this, I have decided to slightly increase my score, with the expectation that the authors will incorporate the revisions they have committed to, such as improving the introduction, refining Section 3, and adding a more comprehensive limitations section.

---

> > > ### Author Response · Authors · 2025-08-04
> > >
> > > We are deeply appreciative of your thoughtful and detailed feedback throughout this entire process. Thank you for taking the time to consider all the reviews and our rebuttal, and for raising your score in light of the discussion.
> > >
> > > We want to confirm that we are fully committed to incorporating all the revisions we promised in the next version of our manuscript. As you noted, we will be focusing on improving the introduction, refining the arguments in Section 3, and adding a more comprehensive limitations section to reflect the valuable feedback we received.
> > >
> > > Thank you again for your constructive engagement, which has significantly helped improve our work.

---

### Official Review · Reviewer_7FBq · 2025-07-05

**Clarity:** 4
**Significance:** 4
**Originality:** 3
**Rating:** 5
**Confidence:** 3

**Summary:**

This work addresses the challenges posed by exploitative pricing (i.e. low pricing from buyers to data annotators where compensation does not reflect skill, effort, and downstream value of contributions) in LLM data markets. Specifically, the authors demonstrate theoretically and empirically that exploitative pricing results in a lose-lose scenario with regards to data sellers being underpaid and data buyers driving sellers away and thereby weakening the data pipeline and limiting model improvements. Based on this, the authors propose a fairshare pricing mechanism that they theoretically show results in win-win scenarios, where sellers maximize profits and buyers achieve higher model performance per dollar spent. Simulation experiments. are used to validate fairshare, and ablations demonstrate the generalizability across different models, tasks, and data valuation methods.

**Questions:**

- The Related Work section (beginning line 106) should include a discussion of Shapley-based data valuation methods.

- 17: LLM --> LLMs

- 49: drive --> driving

-  98: thorugh --> through

**Ethical Concerns:**

["NO or VERY MINOR ethics concerns only"]

**Final Justification:**

I have read the other reviews and rebuttals. I agree with Reviewer AWaS that the idea in the paper is novel and is sufficiently supported, both theoretically and empirically. I will be maintaining my "Accept" rating.

**Limitations:**

Yes

**Quality:**

3

**Strengths And Weaknesses:**

Strengths:
- The work is well-written, theoretically sound, and addresses an important, timely topic.
- As the proposed fairshare pricing mechanism is model and method agnostic, it is flexible as new models and data valuation methods are developed, and flexible to real-world applications desired by potential data-market participants.

Weaknesses:
- Shapley-based data valuation methods are not discussed in the work, nor compared against in the ablation study. Methods such as In-Run Data Shapley (https://openreview.net/pdf?id=HD6bWcj87Y) should make comparison computationally feasible.

---

> ### Author Rebuttal · Authors · 2025-07-31
>
> We thank 7FBq for the thoughtful review and insightful suggestions. We will address your concerns below.
>
> **1. On Shapley-Based Data Valuation Methods:**
>
> We thank the reviewer for this valuable suggestion. This provides a welcome opportunity to demonstrate the generality and modularity of our proposed framework. Our "Fairshare" framework is designed to be agnostic (line 62, 347) to the specific data valuation method used. Its core contribution is to model the market dynamics that result from a set of value scores. To empirically demonstrate this, we ran new experiments during the rebuttal period using the Run-in Shapley-based method and following the same setup as Figure 6 in our paper.
>
> For the MedQA on LLama-3.2-1b-Instruct:
> | Purchased Data Price | BM25 | INFL_IP |DataInf | Run-In Shapley |
> | --- | --- | --- | --- | --- |
> |$500 | 30.2 | 30.6 | 29.4 | 28.3 |
> | $1500 | 30.9 | 31.6 | 29.7 | 31.0 |
>
> As highlighted in these results, the Run-In Shapley method performs similarly with the other data valuation methods. We will include the full suite of results using Run-In Shapley in the next version of our paper.
>
> In addition to this, we will update our Related Work section (Section 2, line 76) to include a thorough discussion of Shapley-based data valuation methods. We will contextualize our chosen methods (e.g., Infl_IP, DataInf) by comparing their scalability and theoretical properties with those of prominent Shapley-based approaches. We will also add this to our Ablation Study discussion (Section 4.3) as a key area for future comparative analysis as these methods become more scalable.
>
> **2. On Minor Typos:**
>
> Thank you for your careful reading of our paper and for catching these errors. We have identified and will correct all the typos identified by the reviewer (line 17: LLM -> LLMs, line 49: drive -> driving, line 98: thorugh -> through) in the revision. We will perform another thorough proofread of the entire manuscript to ensure correctness.
>
> We thank the reviewer once again for their valuable feedback and strong support for our work. We believe the requested changes will make our paper more comprehensive and impactful.

---

> > ### Comment · Reviewer_7FBq · 2025-08-05
> >
> > Thank you for addressing my concerns with the addition of both In-Run Data Shapley experimental results as well as the planned discussion of and contextualization with Shapley-based data valuation methods in the Related Work.
> >
> > I have read the other reviews and rebuttals. I agree with Reviewer AWaS that the idea in the paper is novel and is sufficiently supported, both theoretically and empirically. I will be maintaining my "Accept" rating.

---

> > > ### Author Response · Authors · 2025-08-06
> > >
> > > Thank you very much for your positive feedback and continued support for our paper.
> > >
> > > Your suggestion to engage with Shapley-based methods was a crucial one. We want to confirm that, as promised in our rebuttal, we will be expanding the Related Work section with a thorough discussion of these methods and adding the new experimental results from our Shapley-based comparison to the final version.
> > >
> > > We absolutely agree that these additions have significantly strengthened the paper's claims of generality and robustness. Thank you again for your constructive and insightful review.

---

### Official Review · Reviewer_9cVB · 2025-07-11

**Clarity:** 3
**Significance:** 2
**Originality:** 3
**Rating:** 2
**Confidence:** 4

**Summary:**

Setting:
The paper studies data markets over repeated interactions between persistent sellers (data providers who create and sell data) and a buyer (ML model trainer) in a full information setting i.e. the buyer data valuation is known to all parties. They also define ${p^\ast}_i$ be the maximum price the buyer is willing to pay seller $i$ for their data, taking their budget and marginal utility into account. This price is clearly the best price for the seller - any more and the buyer drops out and any less the seller leaves some money on the table.

Results:
The authors introduce a new assumption about seller behavior - that if their data is priced under the maximum price ${p^\ast}_i$, the seller $i$ is significantly less likely to take part in future markets. The authors show that this implies that consistently pricing under this threshold means that sellers dropout, leading to market collapse. Thus, under the assumptions introduced, it is optimal for buyers as well to price at ${p^\ast}_i$.

**Questions:**

Would similar results hold under a more justifiable seller behavior model as described in weaknesses? Can the authors provide justification for their choice?

**Ethical Concerns:**

["NO or VERY MINOR ethics concerns only"]

**Final Justification:**

The considered seller behavior model is deeply artificial and designed to obtain the results obtained. Here, the sellers essentially pose an ultimatum - either the average price over the runs is specifically $p_j^{\star}$ (which is the max price the buyer would be willing to pay), or they drop off from the market. This was even admitted to as such by the authors in the rebuttal discussion (though they wrongly brush off the seriousness of this flaw). With even a slight perturbation of this assumption, the results and conclusions made collapse.

Further, this is a deeply extractive pricing model, purely benefitting the sellers. It baffles me that the authors chose to call this a "fairshare" pricing.

While I do indeed think this is a very cool problem, and agree with the rest of the reviewers that the paper is well presented, I think the flaw with the seller behavior model is hard to overlook. I suspect the other reviewers who find the results of the paper compelling may not have interrogated the seller response model. I urge them to read the review and discussion posted above.

**Limitations:**

yes

**Quality:**

2

**Strengths And Weaknesses:**

Strengths

- Data valuations are a very important and fast-growing topic. However, past work has mostly ignored the game theoretic aspect. I really like how this work attempts to combine the ML / empirical data valuations (such as influence estimation or attribution techniques) with more classical game theoretic investigations. I see a scope for a lot more such follow up research.
- The repeated interactions model has also been understudied in data markets and, as this paper shows, yields interesting mechanisms and behaviors.

Weaknesses

- The biggest flaw of the work is the assumption about seller behavior - that pricing less than ${p^\ast}_i$ leads to increase in dropout likelihood. This usage of ${p^\ast}_i$ as the threshold used is extremely artificial since ${p^\ast}_i$ has to do with buyer purchase-power and has nothing to do with the seller. In particular, the seller behavior changes given a different set of buyers. A more realistic model would use a buyer-independent threshold based on cost of data creation to the seller. However, in such a model the conclusions shown here would not work.
- Similarly, I contest the naming of ${p^\ast}_i$ to be the "Fairshare" data pricing -- by definition it is the seller-optimal prices. It is unclear how this price is "fair".
- There is also insufficient literature review. In particular, a lot of the work in incentivizing data sharing in federated learning might be highly relevant here. See for e.g. citations in this very recent (parallel) work [[Pang et al. 25](https://arxiv.org/abs/2506.16731)] or [this recent survey](https://www.sciencedirect.com/science/article/abs/pii/B9780443190377000260).
- Finally, the existing work assumes a full information setting where all the participants know the true valuation and budgets of everyone. While this is unlikely in practice, I expect this aspect can be improved upon in future research.


My overall impression is that this is an intriguing but flawed first step along a new research direction.

---

> ### Author Rebuttal · Authors · 2025-07-31
>
> We sincerely thank Reviewer 9cVB for the thoughtful feedback. We appreciate the opportunity to clarify our core assumptions and contributions.
>
> **1. Seller Behavior Assumption and the Role of Cost:**
>
> We thank the reviewer for this crucial question, which allows us to elaborate on our model's core mechanics. We would like to clarify that our choice to model seller behavior around the optimal price ($p^{∗}_{j}$) is a justified one for capturing perceived fairness in a transparent market.
>
> **$p^{∗}_{j}$​ as a Fairness Anchor:** Our framework assumes a transparent market with publicly known data valuation scores (lines 90, 123-125). Here, Maximum Willingness to Pay represents a market-defined anchor reflecting the highest possible value a seller’s dataset $D_j$ can provide to each buyer. Thus, this price also informs **sellers** about the utility their data contributes, not just a buyer-side metric. A rational seller, aware of this maximum, chooses between accepting "satisficing" [10] on any profitable offer (  $\geq c_j$) and optimizing for a fair share of the value created. Foundational literature in behavioral economics, particularly results from the Ultimatum Game [11], demonstrates that agents frequently reject profitable but deeply unfair offers, choosing to punish unfair behavior even at a cost to themselves. This provides strong empirical support for our core assumption: a seller, perceiving a price well below the $p^{∗}_{j}$ anchor as exploitative, may exit the market. Behavioral models of inequity aversion and reciprocity [12,13] further formalize why fairness significantly influences agents' utility and market participation.
>
> This seller reaction to perceived unfairness is the core mechanism that drives the market to a fair, strategic equilibrium. In dynamic settings, sellers exiting the market serve as credible threats of punishment, aligning with the "Folk Theorem" [7] of repeated games. This theorem suggests cooperative outcomes can be sustained through retaliation. Consequently, it becomes the buyer's own optimal long-term strategy to pay the fairshare price to ensure a stable supply of high-quality data. Our theoretical results in Lemma 3 and empirical findings in Section 4.2 confirm this, showing that the buyer's long-term utility is maximized at the same price that maximizes the seller's profit, creating the mutually beneficial "win-win" outcome that we term "Fairshare".
>
> **Integrating Cost as a Foundational Constraint:** We thank the reviewer for highlighting the importance of cost. Our model already incorporates it: as shown in Eqns. (3) and (4) (lines 160–163), sellers must achieve non-negative net profit (revenue minus cost).
>
> In particular, under our current model, the conclusion (Lemmas 1-3) hold as long the optimal price $p^{*}_{j}$ is at least as large as the cost $c_j$.
>
> The reason is that the sellers will still lower their willingness or even refuse to participate in the market when receiving compensation below $p^{*}_{j}$, according to our existing assumption.  Within this viable market, our framework is built on the foundational assumption that sellers behave rationally, seeking to optimize their own welfare (e.g., profit). This principle of the rational, welfare-maximizing agent is the standard and widely adopted axiom that underpins the majority of modern microeconomic and game-theoretic literature [14,15].
>
> On the other hand, if the optimal price $p^{*}_{j}$ is lower than the cost $c_j$, then it means that either the investment of LLM companies or the marginal utility that the data could contribute is lower than the cost to produce these datasets. And this will lead to a market collapse, meaning that there will be no transactions.
>
> However, in the current norm of LLM data market, we believe that in most cases, the companies are still devoting enormous amount of investment to acquire the data as data is still a very valuable asset, which greatly exceeds the cost (e.g., labor cost) to annotate these datasets [17,18]. And this justifies the implicit assumption such that the optimal price $p^{*}_{j}$ is always larger or equal to the cost $c_j$. We thank the reviewer for pointing that out and will update our writing more rigorously in the next version of the manuscript by explicitly adding the assumption regarding the cost and the optimal price.
>
> **Cost-Based Models:** Such a model addresses important questions around production viability, specifically, "Is it profitable to produce this data?" Our current framework, on the other hand, focuses on a distinct, yet equally important, economic phenomenon relevant to ongoing debates about fair compensation and ethical data work: "Am I being compensated fairly relative to the value I create?" Recognizing the significance of both approaches, our work serves as a foundational first step, emphasizing value-based compensation as a means to enhance annotator morale and sustain long-term market participation in the LLM industry. We will clarify this perspective explicitly in our revision and highlight the potential for future work to explore cost-based models in parallel with our approach.
>
> **2. On the Naming of "Fairshare" Pricing:**
>
> We appreciate the feedback regarding the term "Fairshare" pricing. The name was chosen to reflect that our price is not merely seller-optimal, but is a fair equilibrium outcome that benefits both parties in the long run. The fairness of this equilibrium stems directly from our mechanism's core principle: aligning a seller's compensation with the value their data contributes to the buyer's utility, as quantified by data valuation.
>
> As we detailed in our response to the first point, this notion of a sustainable, fair equilibrium is rigorously grounded in both dynamic game theory (e.g., the Folk Theorem [7]) and behavioral economics (e.g., inequity aversion [8,9]). Our theoretical results (Lemma 3) and empirical results (Section 4.2) confirm that this price creates a mutually beneficial "win-win" outcome. The "Fairshare" label captures this crucial alignment of incentives, in contrast to the lose-lose dynamic of exploitative pricing. We will clarify this rationale and its connection to game-theoretic principles in our revision.
>
> **3. Federated Learning (FL) Literature:**
>
> We thank the reviewer for pointing us to this relevant body of work. We are happy to expand our Related Work section to include more references to this parallel research in our revision, including the ones that the reviewer listed. While both our work and the  [Pang et al. 25] tackle data incentivization, they address fundamentally different domains:  [Pang et al. 25] focuses on preventing free-riding in decentralized, collaborative settings, whereas our work pioneers a model for a centralized, transactional data marketplace.
>
> **4. On the Full Information Assumption:**
>
> We thank the reviewer for this critical point. Our choice to begin with a full-information model was a deliberate one, following a long-standing tradition in economic research.
>
> **An Essential Benchmark:**  The full information model serves as an essential benchmark to isolate core strategic interactions and establish a theoretical “best-case” equilibrium. This is crucial for systematically measuring the inefficiencies that arise when more complex forms of information asymmetry are introduced in future work.
>
> **Precedent in Economic Research:** This approach of starting with a complete information model is a hallmark of seminal economic research, like Akerlof's "markets for lemons" [1] and Spence's "job market signaling" [2]. They both used this methodology to establish groundbreaking insights before the field expanded to tackle various forms of asymmetric information. As one of the first papers to formalize the LLM data market with a game-theoretic framework, we believe providing a tractable model is a valuable starting point.
>
> **Scope and Enabling Future Work:** Moving to an incomplete information setting is a vast and important future research direction, not a single extension. It requires extensive assumptions about which party holds private information (cost, budget, utility), their prior beliefs, and the signaling mechanisms available. This opens up a vast research program involving different assumptions about private information and agent beliefs. By establishing the core model, we hope our work enables and encourages future research in this area, as we note in our conclusion (line 357-359).
>
> We will revise our limitations and conclusion sections to better frame our contribution as a tractable starting point for this research area and highlight exciting avenues for future work, such as using Bayesian games to model incomplete information.
> ***
> [1] Akerlof. The market for “lemons”: Quality uncertainty and the market mechanism.
>
> [2] Spence. Job market signaling.
>
> [3] Folger et al. Organizational justice and human resource management.
>
> [4] Folger et al. Fairness theory: Justice as accountability.
>
> [5] Colquitt, et al. What is organizational justice? a historical overview.
>
> [6] Adamovic. Organizational justice research: A review, synthesis, and research agenda.
>
> [7] Fudenberg et al. The Folk Theorem in Repeated Games with Discounting or with Incomplete Information.
>
> [8] Fehr et al. A Theory of Fairness, Competition, and Cooperation.
>
> [9] Rabin. Incorporating Fairness into Game Theory and Economics.
>
> [10] Simon. Rational choice and the structure of the environment.
>
> [11] Güth et al. An experimental analysis of ultimatum bargaining.
>
> [12] Bolton et al. ERC: A Theory of Equity, Reciprocity, and Competition.
>
> [13] Dufwenberg et al. A theory of sequential reciprocity.
>
> [14] Mas-Colell et al. Microeconomic theory.
>
> [15] Von Neumann et al. Theory of games and economic behavior.
>
> [16] Brandom. Scale AI’s Remotasks platform is dropping whole countries without explanation.
>
> [17] Perrigo. Meta’s $15 Billion Scale AI Deal Could Leave Gig Workers Behind.

---

> ### Comment · Reviewer_9cVB · 2025-07-31
> **The economics literature cited seems to *disprove* proposed seller behavior**
>
> Thank you for the detailed replies! I politely disagree with the author's justification and interpretation of relevant economics research in two core questions.
>
> **Artificiality of seller behavior**
>
> > A rational seller, aware of this maximum, chooses between accepting "satisficing" [10] on any profitable offer (
> ) and optimizing for a fair share of the value created. Foundational literature in behavioral economics, particularly results from the Ultimatum Game [11], demonstrates that agents frequently reject profitable but deeply unfair offers, choosing to punish unfair behavior even at a cost to themselves.
>
> This is a great point. I think a model where each seller decides *once* whether to leave proportional to gap to $p^{\star}_j - p_j$ can be justified using the above literature. However, in the proposed model, the seller leave proportional to $p^{\star}_j - p_j$ at *every iteration*. This means that for any $p_j < p^{*}_j$, over a sufficiently long time horizon, the probability of the seller staying is 0.
>
> This is amounts to an **unfair ultimatum by the sellers** - either pay the max price $p^{\star}_j$ or none of us get anything. Note that the buyers have *zero utility* when paying $p^{\star}_j$. And as the response above points out, such an ultimatum is usually rejected.
>
> Please let me know if I misunderstand the dynamics of the proposed marketplace.
>
>
> **What constitutes a fair price?**
>
> Understanding fairness requires understanding how the surplus is distributed amongst the buyers and sellers. Here, all the surplus is taken up by the sellers with buyers having zero utility at the end. Reasonable notions of a "fair" price could equitably split the surplus between the participants, or (better) use the Nash bargaining solution. In fact, there is vast literature in cooperative game theory (including Nash bargaining, Shapley values, envy-freeness) on investigating fariness in surplus division. The current work **contradicts such long-accepted notions of fairness** in market design. This is why I contest to naming this as a "fairshare" pricing.

---

> > ### Author Response · Authors · 2025-08-01
> > **Clarifications on Model Dynamics and Dynamic Fairness**
> >
> > We thank the reviewer for the detailed follow-up, which allows us to clarify these subtle but critical dynamics in our model.
> >
> > **1. Unfair Ultimatum by the Sellers**
> >
> > **Convergence of participation probability:** We would like to gently clarify the mathematical premise that the seller's survival probability converges to zero for any underpayment strategy ($p_t < p^{*}_{t}$) is not universally true.
> >
> > Mathematically, the product of probabilities converges to zero if and only if the sum of the gaps, $\sum(1 - P_t)$, diverges. A strategic buyer could therefore devise a strategy where the underpayment shrinks over time ($p_t → p^{*}_{t}$ sufficiently fast), ensuring the seller's participation probability remains strictly positive.
> >
> > **"Unfair Ultimatum" Interpretation:** We respectively clarify that “either pay the max price pj⋆ or none of us get anything” might not hold since a buyer could **underpay for a finite period** before reverting to the fair price. And the partial participation of the sellers still exists.
> >
> > This highlights our model's core purpose: to frame a rich strategic dilemma for a rational, forward-looking buyer, not a simple seller's ultimatum. The buyer must weigh the short-term gains of various underpayment strategies against the long-term risk of market collapse.
> >
> > Furthermore, your point raises an excellent direction for future work: a "satisfaction range" $[\alpha p*, p*]$ could be modeled where sellers always participate. Crucially, even in such a model, $p^{*}$ remains the essential anchor for this range, and our core conclusion—that value-aligned pricing below this range is self-defeating—would still hold. This is an excellent direction for future work.
> >
> > We will clarify these dynamics and interpretations in our revised manuscript.
> >
> > **Buyer's "Zero Utility/Suplus":** Regarding surplus division, we would like to clarify that the buyer's immediate surplus is zero only in the **specific case** where their budget is not a binding constraint. As defined in our MWP formulation (Equation (9)), our model also explicitly accounts for the common scenario where the budget is the **binding constraint**, in which case the buyer does achieve a **positive immediate surplus**. And in the industry of LLM data market [16,17], this binding scenario is mostly realistic as the actual budgets allocated for data acquisition and annotation labor are often kept exploitatively low.
> >
> > **2. What constitutes a fair price?**
> >
> > We thank the reviewer for this insightful comment, which helps us clarify our focus on a dynamic definition of fairness. We would like to further clarify our model's nuances and its focus on a dynamic and long-term perspective of fairness.
> >
> > The notion of “fairness” is defined differently under different scenarios. Particularly, our work analyzes a dynamic setting where fairness evolves beyond a single-shot surplus split to include **long-term market sustainability**. In our framework, a price is "fair" because it is the unique price that sustains the market and maximizes the **long-term cumulative welfare** for both parties. Any other price that attempts to "split the surplus" more favorably for the buyer in the short term is ultimately unfair to both parties in the long run, as it leads to a lose-lose market collapse. This focus on fairness as a mechanism for sustaining long-term, mutually beneficial relationships is well-supported by a vast literature on different game structures.
> >
> > Crucially, our central argument—that unfair compensation damages long-term relationships—is a robust finding supported. Foundational work on relational contracts [1] and experimental results from repeated bargaining games [2] both show that the value of future cooperation is precisely what sustains fair behavior in the present.
> >
> > The reviewer's suggestion to use a Nash Bargaining Solution is excellent. Such a model would focus on non-sequential, cooperative bargaining outcomes rather than the non-cooperative, price-setting sequential dynamics of our Stackelberg game. We chose the latter for this setting because we believe the LLM data market moves in a more sequential manner.
> >
> > We agree a bargaining model is a valuable direction for future work. And our framework serves as a crucial precursor of the bargaining game by providing a transparent guide to $p^*$. This both empowers exploited sellers with a clear estimate of their contribution and secures the buyer's long-term welfare by identifying the price that sustains the market. This explicit price thus serves as a foundational layer for any bargaining game by providing a well-justified prior for the bargaining room.
> >
> > In sum, we chose the term "Fairshare" to describe this **dynamically fair, sustainable, win-win equilibrium price**, to distinguish it from static surplus-splitting concepts. We will clarify this distinction in our revision.
> >
> > [1] Baker et al. Relational Contracts and the Theory of the Firm.
> >
> > [2] Rubinstein. Perfect Equilibrium in a Bargaining Model.

---

> > > ### Comment · Reviewer_9cVB · 2025-08-01
> > >
> > > Thank you for the response.
> > >
> > > The response above seems to agree to the core argument that the sellers pose an ultimatum of either the "underpayment shrinks over time ($p_t → p^{*}_{t}$) sufficiently fast" or none of us get anything.
> > >
> > > Further, for analyzing the fairness, the core argument seems to be:
> > > > Any other price that attempts to "split the surplus" more favorably for the buyer in the short term is ultimately unfair to both parties in the long run, as it leads to a lose-lose market collapse
> > >
> > > This collapse is a direct result of the assumed artificial seller response as stated. Not an actual property of any realistic marketplace. I again urge the authors to look into cooperative game theory concepts for fair pricing notions.

---

> > > > ### Author Response · Authors · 2025-08-04
> > > > **Justification on Our Methodological Framework**
> > > >
> > > > We thank the reviewer for the follow-up. The core discussion now concerns the most appropriate framework for modeling fairness in the LLM data market (e.g., cooperative vs. non-cooperative games). We clarify our choice of a **non-cooperative, behavioral** framework.
> > > >
> > > > **1. The Buyer's Incentive to Accept a "Fairshare" Price**
> > > >
> > > > We would like to respectfully clarify a core misinterpretation:(1) our model is not about a “seller’s ultimatum but rather a **buyer's rational, long-term trade-off**, and (2) the buyer has a **strong incentive** to pay the "Fairshare" price. The reviewer's argument "left with nothing" applies **only** to cases where the buyer’s budget exceeds the data’s value.
> > > >
> > > > Our model is more general, also covering budget-constrained cases where buyers still achieve a positive surplus. This is especially relevant in today’s LLM market, where data’s economic return often far exceeds exploitatively low annotation budgets [1–3].
> > > >
> > > > In either case, a rational buyer seeks to maximize **cumulative welfare in the long term**, a principle from dynamic programming, accepting lower short-term surplus to sustain data supply and maximize long-run gains. This rational choice, not a submission to a threat, is our model's central dynamic.
> > > >
> > > > **2. The Study of Fairness in Cooperative and Non-Cooperative Games**
> > > >
> > > > We wish to clarify that fairness is **central in both cooperative and non-cooperative games**, though analyzed differently.
> > > >
> > > > **Cooperative games** typically define fairness axiomatically, through solution concepts (e.g., Nash Bargaining) that prescribe a fair outcome for players who can form binding agreements [13].
> > > >
> > > > **Non-cooperative games** analyze fairness **behaviorally**. This approach, a cornerstone of behavioral game theory, incorporates a preference for fairness directly into players' utility functions to study how these preferences shape strategic equilibria. This is formalized in canonical models of Inequity Aversion [4, 6] (where agents dislike unequal outcomes) and Reciprocity [5, 7] (where agents punish unfair actions).
> > > >
> > > > Our paper sits in this rich, non-cooperative tradition. Our Assumption 1—that sellers exit in response to exploitative prices—is a classic behavioral assumption designed to study how such fairness preferences affect the long-term sustainability of a market.
> > > >
> > > > **3. Why a Non-Cooperative Model Fits the LLM Data Market**
> > > >
> > > > More fundamentally, the choice between a cooperative and non-cooperative model must be grounded in the real-world characteristics [16].
> > > >
> > > > **Cooperative models** suit settings where a few dominant players are locked into a negotiation and must form a binding agreement to divide a joint surplus(e.g., a formal business merger or a single, critical supplier negotiating with a single buyer)  [13,16].
> > > >
> > > > **Non-cooperative models**, in contrast, are suited for decentralized markets with many independent, self-interested transactions where binding grand coalitions are not feasible. This includes auctions, price competition between firms [13,16], and, we argue, the **LLM data market**, especially on gig-work platforms [12].
> > > >
> > > > Given the LLM data market is a landscape of millions of individual, sequential transactions, our choice of a non-cooperative Stackelberg game is a deliberate one to capture these real-world dynamics. It allows us to answer our specific research question: How do the self-interested actions of individual buyers and sellers, who are sensitive to fairness, affect the long-term sustainability of the market? A cooperative model, while valuable, would answer a different question about an idealized grand bargain.
> > > >
> > > > This does not mean we ignore fairness; we analyze it **behaviorally** within our non-cooperative framework. Our model shows how a sustainable outcome can emerge from self-interested play precisely because agents have fairness preferences, a principle rigorously supported by literature on relational contracts and repeated bargaining games.
> > > >
> > > > We thank the reviewer for the thoughtful suggestions and will add a discussion on extending our work to cooperative frameworks in our future directions.
> > > > ***
> > > > [1] Niamh. Millions of Workers Are Training AI Models for Pennies.
> > > >
> > > > [2] Tan et al. Behind the AI boom, an army of overseas workers in ‘digital sweatshops’.
> > > >
> > > > [3] Brandom. Scale AI’s Remotasks platform is dropping whole countries without explanation.
> > > >
> > > > [4] Bolton et al. ERC: A Theory of Equity, Reciprocity, and Competition.
> > > >
> > > > [5] Dufwenberg et al. A theory of sequential reciprocity.
> > > >
> > > > [6] Fehr et al. A Theory of Fairness, Competition, and Cooperation.
> > > >
> > > > [7] Rabin. Incorporating Fairness into Game Theory and Economics.
> > > >
> > > > [12] Gray et al. Ghost work: How to stop Silicon Valley from building a new global underclass.
> > > >
> > > > [13] Osborne et al. A Course in Game Theory.
> > > >
> > > > [14] Rabin. Incorporating Fairness into Game Theory and Economics.
> > > >
> > > > [15] Stokey et al. Recursive Methods in Economic Dynamics.
> > > >
> > > > [16] Tirole. The Theory of Industrial Organization.

---

### Note · Authors · 2025-08-13

We thank the AC and reviewers for an engaged discussion. First, we appreciate reviewers' recognition of a novel, scalable idea (AWaS) on a timely topic (9cVB, 7FBq), combining ML data valuation with game-theoretic analysis (9cVB). Reviewers also noted well-supported results (AWaS, nL1m), clear writing (AWaS), and policy/fairness impact (nL1m).

The rebuttal process allowed us to address concerns from nL1m, AWaS, and 7FBq:
- **Strengthened Theoretical Guarantees:** tightened core assumptions—seller participation, explicit role of cost, and our non-cooperative modeling choice tied to observed market dynamics.
- **Enhanced Empirical Validation:** added robustness experiments, including a Shapley-based data-valuation method and non-affine utility mapping.
- **Improved Clarity, Motivation, and Scope:** clarified links to real-world practice, added figure/notation fixes, and expanded limitations to foreground future work on incomplete information settings.

The discussion with reviewer 9cVB centered on (i) cooperative vs. non-cooperative modeling and (ii) definitions of fairness. Our stance:
- **Non-cooperative, behavioral analysis** is adopted to match the LLM data market’s reality: millions of independent, sequential transactions. This lens—standard in behavioral economics—captures how fairness preferences shape equilibrium behavior [1,2].  A cooperative model would answer a different question about an idealized grand bargain among dominant buyer and seller and is a valuable future research direction.
- **Dynamic definition of fairness.** We call the optimal price “fair” because it sustains participation and maximizes long-run welfare for both sides—consistent with relational-contract and repeated-game results.
- **Buyer rationality, not ultimatum.** A forward-looking buyer makes strategic moves by paying the fairshare price because it is the only strategy that prevents market collapse and maximizes their overall gains over time.

Ensuring data markets are fair, transparent, and sustainable is critical for AI. We see this paper as a first step toward formalizing LLM data market dynamics via data valuation. We will incorporate the promised revisions (theoretical clarifications, robustness results, expanded limitations) and hope this catalyzes further research on incomplete information and alternative bargaining structures.

[1] Fehr et al. A Theory of Fairness, Competition, and Cooperation.

[2] Rabin. Incorporating Fairness into Game Theory and Economics.

---

### Decision · Program_Chairs · 2025-09-17

**Decision:**

Accept (poster)

**Comment:**

The review team is largely supportive of the paper, with three strong supporters with scores of 5. Reviewers praised both the formalization/applied modelling aspects of the paper when it comes to markets for data involving LLMs, as well as the novel proposed fairshare pricing mechanism. Overall, the contribution is clearly novel, timely, and impactful.

The one negative reviewer notes that they would like to see the problem studied through the lens of cooperative game theory. I agree with the reviewer that the paper, should, for the camera-ready, discuss the connection to cooperative game theory and explain the relationship to this line of work. However, I do fully agree with the authors' response that this is only one of possibly several possible framings, note that the authors provide compelling justification in their response for their non-cooperative framing, and that this critique is, in my opinion, insufficient grounds for rejection. I will note that the reviewer did not engage with the rest of the review team nor tried to adjust their score based on the authors' response, the other reviews, and the discussion where they were explicitly prompted to further support their disagreement with the rest of the review team.

As such, I recommend the paper for acceptance as a poster.